# The interplay between social dominance and decision-making under expected and unexpected uncertainty: Evidence from event-related potentials

Saeedeh Khosravi[1], Lydia Kogler[2,3], Reza Khosrowabadi[4], Touraj Hashemi[5], Birgit Derntl[2,3]*, Soomaayeh Heysieattalab[1]*

1 Department of Cognitive Neuroscience, University of Tabriz, Tabriz, Iran, 2 Department of Psychiatry and Psychotherapy, Women's Mental Health & Brain Function, Tübingen Center for Mental Health (TüCMH), University of Tübingen, Tübingen, Germany, 3 German Center for Mental Health (DZPG), partner site, Tübingen, Germany, 4 Institute for Cognitive and Brain Sciences, Shahid Beheshti University, Tehran, Iran, 5 Department of Psychology, Faculty of Education and Psychology, University of Tabriz, Iran

* heysieattalab@gmail.com, heysieattalab@tabrizu.ac.ir (SH); Birgit.Derntl@med.uni-tuebingen.de (BD)

## Abstract

Decision-making is a fundamental aspect of human behavior, especially in uncertain situations where social interactions play a significant role. Social dominance, which involves power dynamics within groups, holds the potential to shape decision-making. Individuals' expectations and certainty about outcomes are crucial for monitoring their performance in social dominance situations. However, the impact of expected and unexpected uncertainty on decision-making in social dominance contexts remains unclear. This study aimed to unravel the neural and behavioral patterns associated with decision-making across varying social dominance levels under conditions of uncertainty. Researchers investigated this by analyzing brain activity in 51 students. Participants were presented with both positive and negative feedback under conditions of both expectation and uncertainty, while their brain activity was recorded using electroencephalography (EEG). Specifically, we investigated the properties of key neural correlates of feedback processing, including feedback-related negativity (FRN), and P3 components of event-related potential (ERP), and reward prediction error (RPE) signals. The results revealed that the low-dominance group exhibited a larger FRN amplitude than the high-dominance group. Also, unexpected-uncertain negative feedback elicits a stronger FRN amplitude than other conditions. P3 amplitude was larger for high-dominance compared to low-dominance individuals. Additionally, P3 amplitude varied by feedback valence and condition, with larger amplitudes for positive feedback and unexpected-uncertain conditions. In FRN wave difference, the high-dominance individuals exhibited more negative amplitude in unexpected-uncertain conditions. This reveals distinct neural responses to uncertainty and feedback between individuals with high and low dominance, suggesting

**Data availability statement:** All the authors agreed that the data supporting the findings of this study need to be openly available and accessible on the Open Science Framework (OSF) platform. We have agreed to this method of data sharing in the first stages of this study. The dataset is assigned a persistent identifier and can be accessed directly at the following link: https://osf.io/4rsmq/.

**Funding:** This project holds a scholarship entitled "TabrizU-300" by the International Academic Cooperation Directorate University of Tabriz, Iran. The funders had no role in study design, data collection and analysis, decision to publish, or preparation of the manuscript.

**Competing interests:** No conflicts of interest were declared.

that social hierarchy modulates brain mechanisms underlying decision-making and reward processing.

## 1. Introduction

Humans face a tradeoff between exploration and exploitation when making decisions, which influences behavior from daily activities to occupational choices [1]. Decision-making is a complex and challenging process, and the outcomes of decisions are often unclear and may lead to unfavorable results, requiring individuals to be mindful of uncertainty while making choices [2,3]. Uncertainty in decision-making comes in two forms: expected and unexpected. Expected uncertainty is the familiar unreliability of actions with probabilistic outcomes, like flipping a coin. Unexpected uncertainty, on the other hand, is a surprise – a sudden change in a previously reliable association, like a lever that always gave a reward but now doesn't. In other words, unexpected uncertainty arises from a violation of strong action-outcome links, while expected uncertainty stems from violations of weak, probabilistic ones [4].

In contexts characterized by uncertainty, efficient decision-making is crucial to minimize harm and optimize well-being and prompt assessment of the consequences of actions holds significant importance [5]. Individuals heavily rely on feedback—whether positive or negative—to guide their future behavior. The brain likely employs specialized mechanisms to assess outcome value, magnitude, and to link feedback information with mental importance and motivation [6]. This cognitive process is facilitated by distinct systems in the brain dedicated to processing rewards and losses [5].

Theoretical frameworks pertaining to reinforcement learning (RL) furnish a valuable paradigm for comprehending these neural mechanisms. RL theory asserts that learning occurs through iterative experimentation, wherein individuals assess the repercussions of their behaviors—actions yielding outcomes that exceed expectations are reinforced, whereas those resulting in outcomes that fall short of expectations are diminished. Central to this process is the concept of the prediction error (PE), which is the discrepancy between expected and actual outcomes, driving internal learning signals and expectation updates [7,8]. One of the electroencephalographic components, known as the feedback-related negativity (FRN), is a well-established neural marker associated with loss processing [9]. This negative deflection is typically observed 200–300 milliseconds post-feedback onset in the frontal region, exhibits larger amplitudes in response to negative feedback compared to positive feedback [7,9–16]. Beyond simple outcome valence, the FRN is sensitive to factors such as uncertainty and unexpectedness [12]. Research suggests that the FRN reflects the encoding of PEs, aligning with RL theory [16].

Within the RL framework, the FRN is understood to reflect a rapid negative PE signal—often tied to mesencephalic dopamine signals—relayed to the anterior cingulate cortex (ACC), where it supports behavioral adaptation by identifying outcomes that are worse than predicted. The salience of this signal increases when outcomes are surprising or when uncertainty is high, making FRN a core index of adaptive learning

[7,17]. Studies by Yu et al. (2011) [18] and Pfabigan et al. (2011) [12] have shown that the FRN range is modulated by these factors, particularly for unexpected and uncertain outcomes. This suggests that the FRN may categorize outcomes as "good" or "bad" and signal "worse-than-expected" results via reward prediction error (RPE) signals. RPE represents the discrepancy between expected and actual outcomes, driving internal expectation updates [11,19–23]. The amplitude of RPE signals is influenced by outcome expectation, determined by factors such as certainty and likelihood. Mushtaq et al. (2011) [24] define certainty as the ability to accurately predict future events.

The P3 component plays a significant role in outcome evaluation and reward processing [25,26]. Following the detection of a mismatch between expected and actual results, the prediction model is updated to enhance its accuracy for future feedback. In response to reward and punishment stimuli, a positive deflection, known as P3, appears between 300 to –500 milliseconds post-feedback and is linked to this mechanism [27]. Beyond its initial sensitivity to stimulus meaning and probability, the P3 component of the event-related potential (ERP) is essential in higher-order cognitive processes. It contributes to decision-making [27] and outcome evaluation [13], dynamically adapting its response to assess the functional significance of feedback stimuli [10,28]. This goes beyond mere performance tracking, reflecting the subjective significance and surprise associated with the stimuli [12,29].

In RL terms, the P3 component is thought to reflect the allocation of attentional resources and motivational significance to feedback—particularly rewarding feedback. While the FRN signals an initial, automatic PE, the P3 is more involved in the consolidation and evaluation of the feedback's affective and motivational impact. Together, these components index distinct but complementary aspects of the RL feedback loop: immediate error detection (FRN) and the higher-order appraisal of outcome relevance (P3) [7,17].

Furthermore, the P3 component demonstrates responsiveness to the absolute magnitude of rewards irrespective of the valence of the outcomes (positive or negative). This suggests a potential limitation in the evaluative function of feedback negativity, possibly restricting it to classifying events as either highly or lowly rewarding [30]. Although examining brain activity provides insights into individual decision-making under uncertainty, a comprehensive understanding requires acknowledging various influencing factors. These include psychological aspects (e.g., biases and preferences) [31–33], and situational influences (e.g., social context and available information) [34–37]. Social status, for example, can affect decision-making under both certain and uncertain conditions [37], with individuals sometimes altering their decisions when in conflicting social groups, whether consciously or unconsciously [38].

Importantly, recent theoretical advancements propose that RL processes transcend mere individual decision-making and may elucidate how individuals maneuver through intricate social landscapes, encompassing the formation and reinforcement of social dominance hierarchies. In this framework, dominance can be interpreted as an emergent characteristic of reward-oriented learning—individuals recalibrate their behavioral approaches based on perceived social rewards and penalties, acquiring knowledge regarding which behaviors engender deference or influence [7,39].

Acquiring a dominant position within a hierarchy not only offers advantages but also facilitates the formation of effective social alliances by clearly indicating dominant relationships [40–42]. Human social interactions are often marked by competition for hierarchical superiority. The pursuit of high status is conceptualized as dominance, a trait-like individual difference in personality and social psychology. Individuals vary in their propensity for dominance, with those showing higher levels more likely to achieve desired high-status positions [41]. Behaviorally, dominance manifests as coercive control over resources, enabling individuals to command deference. Empirical evidence underscores the importance of dominance, distinguishing it from prestige, which relies on knowledge or shared benefits. Dominant individuals leverage their coercive power to gain social influence and preferential access to resources [43,44].

Social dominance, signifying an individual's relative power and hierarchical position, exerts a demonstrable influence on decision-making processes. Individuals high in social dominance exhibit faster decision-making responses across various tasks, indicating a predisposition towards promptness in decision-making regardless of social context. Such activity has been associated with heightened brain function in regions including the insula, cingulate, right inferior temporal gyrus, and

right angular gyrus [44]. Moreover, individuals with elevated social status tend to prioritize rewarding outcomes, whereas those with lower status tend to assess outcomes in terms of potential risks and threats [45,46].

While prior research on dominance mainly focused on behaviors within competitive social contexts, little is known about underlying individual traits [47,48]. This study posits that specific, context-independent traits of dominant individuals contribute to their social influence. Notably, unlike previous works that investigate how individual traits or manipulations modulate feedback processing [12,18,22], this study explores how social dominance levels lead to distinct neural and behavioral patterns for both positive and negative feedback across varying uncertainty levels. Therefore, our work aims to highlight how core feedback mechanisms—indexed by FRN and P3—are differentially modulated by stable personality traits like dominance, especially under varying levels of uncertainty. This provides new insights into how individuals learn to navigate complex, socially embedded environments using biologically grounded PE signals.

The main question of this study was whether the amplitude and latency of the FRN and P3 components differ between high-dominance and low-dominance individuals when decision outcomes are either expected or unexpected. Using a well-established decision-making paradigm [12,22], participants received positive and negative feedback under three feedback conditions—expected-certain, expected-uncertain, and unexpected-uncertain—while EEG recordings captured FRN and P3 activity. Based on prior research indicating that individuals with lower social dominance are more attuned to social evaluation and exhibit heightened sensitivity to external judgments in uncertain or hierarchical contexts [46,49], we hypothesized that they would show more pronounced behavioral responses to negative feedback—particularly when outcomes are unexpected and uncertain—as a reflection of increased performance monitoring and concern for evaluation. At the neural level, we predicted that low-dominance individuals would exhibit larger FRN amplitudes in response to negative feedback under unexpected-uncertain conditions, indicating greater sensitivity to negative PEs. In contrast, we expected P3 amplitudes to be generally higher in response to unexpected-uncertain feedback than expected feedback across all participants, with high-dominance individuals showing larger P3 amplitudes overall due to their stronger motivational engagement and greater sensitivity to reward salience. Furthermore, to isolate neural correlates of RPE, we computed valence difference waves (negative minus positive feedback) for both FRN and P3 components [12,22,50–54]. Given the evidence that high-dominance individuals exhibit stronger responses to reward cues and are more driven by reward-seeking behavior [49,55–59], we expected more pronounced negative RPE signals in the unexpected-uncertain condition among high-dominance individuals relative to their low-dominance counterparts.

## 2. Material and method

### 2.1. Participants

A total of 51 healthy adult participants (male: 22, female: 29, range: 18–45 years (mode: 23–27 years)) participated in this study (Based on a priori power analysis conducted with G*Power software [60–62] with a medium Cohen's F (0.20), power of 0.95, and an alpha error probability of 0.05, the minimum required sample size for this study was 44 participants). Participants were recruited from Tabriz University, Iran, between 22/06/2021 and 23/11/2021. The initial sample (Fig 1) included 400 students who completed the personality research form dominance subscale (PRF-d [63]), Spielberger's state-trait anxiety inventory (STAI-T) [64], and a standardized handedness questionnaire [65]. From the 400 students, 21 left-handed participants (10 females), and 8 participants with a history of neurological or mental disorder were excluded. Further, to account for the intricate relationship between trait anxiety and social dominance, and the potential for anxiety to influence both social dominance and behavioral outcomes, we employed the STAI [64] to control this factor. Participants scoring above the established high-anxiety threshold of 53 [66,67] were excluded from the study. Following this approach, we had to exclude 17 participants [10 females (M = 61.90, SD = 7.38), and 7 males (M = 69.14, SD = 2.84)]. Following, 32 participants (17 females) who had exactly obtained the cut-off score of the questionnaire (PRF-d = 9 [44,63]) were excluded.

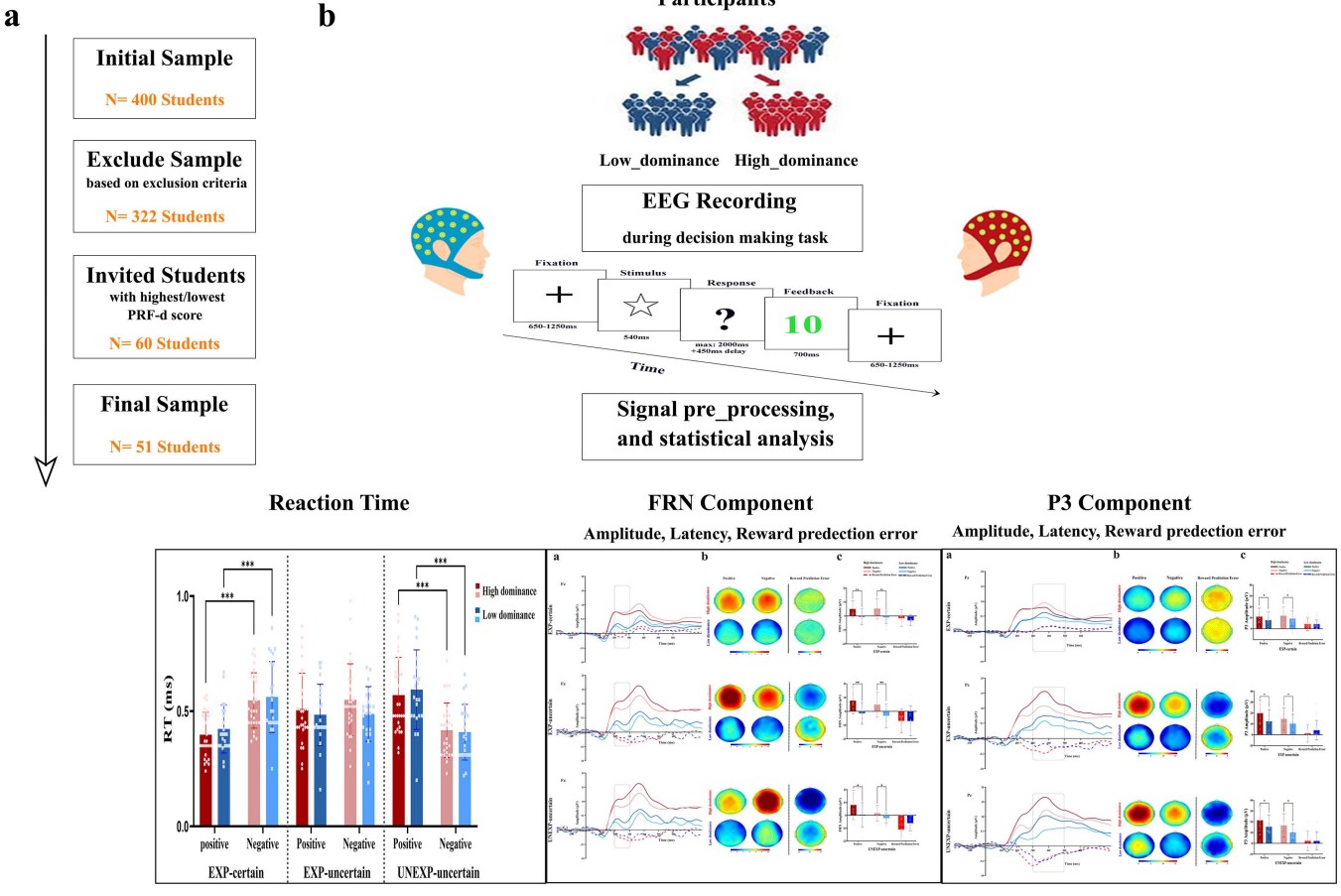

**Fig 1. Study design.** a) Participant selection procedure, b) Experimental procedure.

The following, the remaining 322 participants' scores were rank ordered and were divided based on their scores in the PRF-d into two groups of low and high dominance (S1 File) [44,63]. We then selected the 30 lowest scorers (low-dominance group) and the 30 highest scorers (high-dominance group). Although the PRF-d scale theoretically ranges from 0 to 16, our final groups did not include participants at those absolute extremes; to reach our target sample size while preserving clear separation, we included scores within a small range around each endpoint_ one standard deviation above or below the cutoff. These 60 participants were subsequently invited to take part in the EEG experiment. While 5 participants declined further participation (4 low dominance, 1 high dominance), 55 participants were recruited for the EEG experiment. Two participants (low dominance), however, couldn't withstand the cap and experimental circumstances until the end and withdrew during the EEG recording. Due to software problems, two additional participants (1 low dominance, 1 high dominance) were removed since their EEG signals could not be accessed, and an error occurred. Thus, the final sample consists of 51 participants [(low-dominance group: n = 23 (female = 12); mean score PRF-d = 5.35 (±1.36), (high-dominance: n = 28 (female = 17); mean score PRF-d = 12.64 (±1.47)]. All final samples of 51 participants were right-handed and had normal or corrected-to-normal vision, with no reported history of neurological or psychiatric disorders. Before the experiment, the participants gave their written consent after receiving detailed written information. The Ethics Committee of the University of Tabriz approved the study and all experimental protocols (IR. TABRIZU.REC.1400.011).

## 2.2. Instruments

### 2.2.1 Personality measurements.

The personality research form dominance subscale (PRF-d [63]) was used to evaluate the social dominance motivation of the participants. It was filled out via an online survey (www.docs.google.com/forms) a few days before the experiment. This subscale has 16 true/false items (e.g., "I feel confident when directing the activities of others/I have little interest in leading others.") that make positive and negative assertions about the motive for social dominance [44]. Each true statement related to dominance earns us one point, but we need to pay attention to the reverse statements. For instance, half of the questions (e.g., I try to control others rather than permit them to control me) are formulated in the way that a "true" response earns one point, and a "false" answer earns zero points. However, the other half of the questions are the opposite (e.g., I would make a poor military leader), where a "false" answer earns one point, and a "true" answer earns zero points. Hence, the minimum score that can be achieved was zero, while the maximum score was 16 [63].

### 2.2.2 Decision-making task.

To investigate the impact of dominant on decision-making we relied on a modified version of the gambling paradigm presented by Kogler et al. (2017) [22], and Pfabigan et al. (2011) [12]. The task included training and experiment sessions. First, a training session consisting of 60 trials was carried out in which the participants were asked to explore and learn particular cue-response mappings. Each trial began with a fixed point on the center of the screen that lasted randomly for 650–1250ms. Then, the geometrical shape (star, circle, or triangle: height and width: 2.5 cm) was displayed as the imperative cue for 540ms. Participants were then instructed to respond within 2000ms by pressing a designated button upon seeing a question mark. Finally, they received feedback for 700ms in which the green 10 indicated correct responses and a 10,000-Rial (the common currency of the country) reward, while the red 10 indicated wrong answers and a 10,000-Rial loss.

During training, the three shapes were linked to two buttons in a specific manner (Table 1): The first shape (star) guaranteed a reward (100% chance) and was solely associated with the first button. The second shape (circle) was connected only to the second button and offered a 50% chance of a reward when pressing the second button. The third shape (triangle), regardless of the button pressed, resulted in no reward (0%). Thus, the participants were supposed to fulfill expectations concerning the relationships between the cue shapes and the results of the provided feedback according to the certain methods specified for the first and third shapes (100% positive result for the first shape after pressing key 1, 100% negative result for the third shape regardless of what key was pressed) and the uncertain and probabilistic method

**Table 1. Reward probability, Number (Num) of trials, Expectancy, and valence per cue in training and experimental sessions. We also list the mean (SD) of the number of artifact-corrected trials for high and low dominance (dom.).**

| Training | | | | | Experimental | | | |
|---|---|---|---|---|---|---|---|---|
| | Reward probability | Num. of trials | Feedback-type condition | | Reward probability | Num. of trials | Num. of artifact-corrected trials | |
| | | | Valence | Expectancy | | | High dom. | Low dom. |
| **Cue1 (Button 1)** | | | | | | | | |
| | 100% | 20 | Positive | EXP-certain | 80% | 170 | 156.68 (19.84) | 153.43 (21.16) |
| | | 0 | Negative | UNEXP-uncertain | | 40 | 32.43 (6.28) | 35.96 (9.53) |
| **Cue2 (Button 2)** | | | | | | | | |
| | 50% | 10 | Positive | EXP-uncertain | 50% | 40 | 36.50 (19.19) | 34.48 (21.15) |
| | | 10 | Negative | EXP-uncertain | | 40 | 34.21 (14.53) | 34.48 (14.97) |
| **Cue3 (Button 1/2)** | | | | | | | | |
| | 0% | 0 | Positive | UNEXP-uncertain | 20% | 40 | 35.14 (5.87) | 36.74 (5.32) |
| | | 20 | Negative | EXP-certain | | 170 | 155.86 (16.60) | 162.04 (16.71) |

specified for the second shape (50% chance of positive or negative outcome by pressing the second button). At the end of the training session, participants were asked about their subjective estimates of cue-feedback contingencies.

Then, during the experimental session, the participants were asked to collect as much money as possible (based on 20,000-Rial deposits) based on the attempt-result-action relationships they had already mastered. However, the probability of getting awarded was modified in this stage: the first shape was now associated with an 80% chance of getting awarded by pressing the first button, and the third shape with a 20% chance of getting awarded regardless of the selected button. Thus, the absolute expectation that had been acquired during the previous stage was violated (20% unexpected negative feedback for the first shape and 20% uncertain positive feedback for the third shape). Nevertheless, in 80% of the trials, the examinees continued to receive the expected feedback (80% positive-expected feedback for the first shape and 80% negative-expected feedback for the third shape). The probability of the second shape (the second button, 50%) did not change. Consequently, the likely feedback was not violated (50% unexpected positive and negative feedback). Using the above task, the experimental scheme examined the following three levels of expectation: 1) expected-certain feedback, 2) expected-uncertain feedback, and 3) unexpected-uncertain feedback, which all had positive or negative values (Fig 2, Table 1).

In the experimental session, 500 trials were introduced. After each 50-trial block, the participants were provided with general performance feedback and were given a short rest. At the end of the experiment, the participants were asked to estimate the probability of rewards for the three shapes for the training and experimental sessions to assess if the experimental manipulation was successful. The entire experiment lasted approximately 60 minutes.

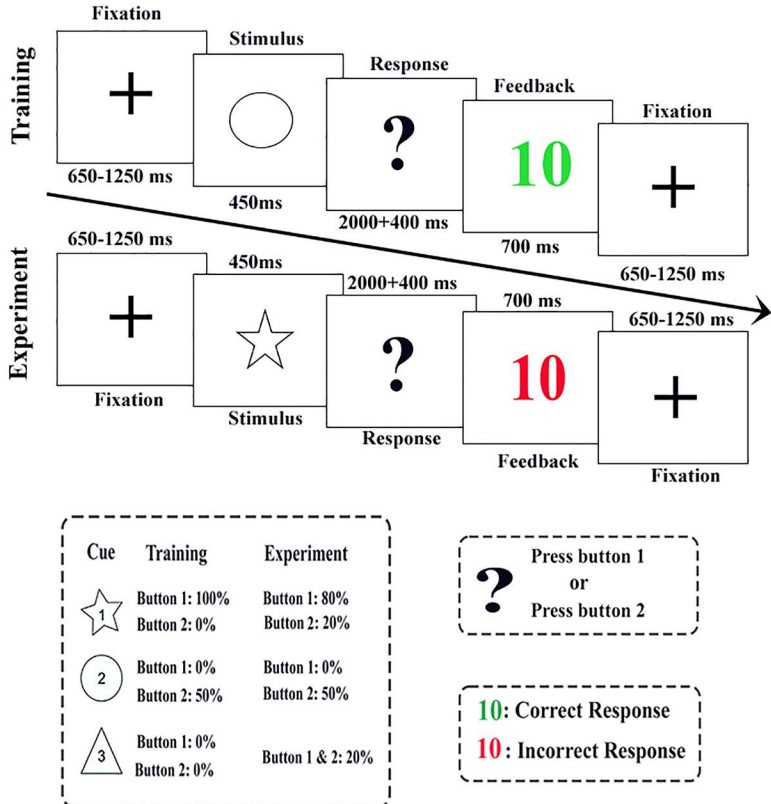

**Fig 2. Schematic representation of the trials used in the decision-making task.**

The paradigm was piloted with three participants to confirm that the cue-response mappings could be learned within the allotted number of trials. The behavioral paradigm was programmed with MATLAB R2011a. During the experiment, participants sat comfortably with their heads roughly 50 centimeters from the computer screen.

### 2.3 EEG recording and signal pre-processing

Continuous electroencephalogram (EEG) signals were captured using a 64-channel waveguard™ cap (ANT Neuro, Enschede, Netherlands) following the 10–10 system configuration. A mastoid reference electrode was connected, and the ground electrode was placed on the Afz electrode. A band-pass filter with a range of 0.3–50 Hz was applied to all signals, which were then digitized at a sampling rate of 250 Hz. To ensure optimal signal quality, electrode impedance was maintained below 10KΩ. Participants were asked to remain as still as possible throughout the recording process.

Using EEGLAB 2021.1, the raw EEG signals were first filtered with a windowed FIR sync filter to remove line noise and other artifacts by applying a high-pass filter to frequencies above 0.3 Hz and a low-pass filter to frequencies below 30 Hz. Next, the EEG recordings were inspected visually to detect artifacts. Epochs with artifacts were thrown out and not used for further processing. Eye blinks, muscular tension and lousy channel artifacts were identified and removed through independent component analysis (ICA) (For each participant, the percentage of epochs that were rejected is below 10%.). Finally, no channels were identified as consistently defective, and therefore no electrodes required interpolation, and EEG signals were prepared for ERP processing. Following that, a semi-automated method for identifying artifacts was utilized to eliminate epochs that displayed voltage values surpassing ±75 µV or voltage drifts greater than 50 µV across all EEG electrodes (Table 1 shows the exact number of artifact-correct trials; the low- and high-dominance groups had no significant differences in any condition (all $T < 1.32$, $p > 0.136$)).

Epochs were extracted from 200ms before feedback onset to 700ms after it. The average of the first 200ms was considered as the baseline interval. Grand averages of the six conditions were calculated after averaging artifact-free trials per participant and condition. Trials were grouped depending on expectation, certainty, and valence. Thus, there were six different conditions: (i) expected-certain positive feedback (Cue 'one'; EXP-certain (POS)), (ii) expected-certain negative feedback (Cue 'three'; EXP-certain (NEG)), (iii) unexpected-uncertain positive feedback (Cue 'three'; UNEXP-uncertain (POS)), (iv) unexpected-uncertain negative feedback (Cue 'one'; UNEXP-uncertain (NEG)), (v) expected-uncertain positive feedback (Cue 'two'; EXP-uncertain (POS)), and (vi) expected-uncertain negative feedback (Cue 'two'; EXP-uncertain (NEG)). Subsequently, the average number of positive from the average number of negative feedback trials for each participant were subtracted to measure RPE-related brain activity across the three levels of expectation that lead to EXP-certain, EXP-uncertain, and UNEXP-uncertain conditions [12,22].

For the quantification of the FRN and P3 components, mean amplitude values were assessed 200–300ms after feedback onset at three electrodes (Fz, FCz, and Cz) for FRN amplitude and 300–500ms after feedback onset at Pz for P3 amplitude. The selection of these time windows and electrode locations was based on the literature revealing that the frontocentral electrodes show greater FRN amplitudes [10,12,14], and P3 amplitude increases from frontal to parietal region [13,68]. Further, latencies from feedback onset to the original waveforms' minimum (FRN) and maximum (P3) were also extracted.

### 2.4. Statistical analysis

To test for normal distribution of our data, the Kolmogorov-Smirnov test was conducted. Parametric tests were employed when the data satisfied the normality assumption. In cases where normality was violated, non-parametric tests were run instead. In the results section, we explicitly state when a parametric vs. non-parametric test was applied.

The data were evaluated in two parts: behavioral measures (subjective estimates, positive feedback, and reaction times (RTs; measured from the stimulus onset to the response onset)) and ERP results.

For behavioral measures, the responses of low- and high-dominance groups to different cues during training were compared using the Mann-Whitney U test, and differences in the priority of the button click were assessed with the Wilcoxon test. Finally, a mixed-design repeated measures ANOVA was used to examine the effects of experimental manipulation on RT, with the between-subject factors dominance (high vs. low dominance), and the within-subject factors condition (EXP-certain vs. EXP-uncertain vs. UNEXP-uncertain) and valence (positive vs. negative) was used to examine the effects of experimental manipulation on RTs. For more details of behavioral effect and the evaluation of the trial (or block) effect over time (as indicative of a learning effect), we employed linear regression analyses on accuracy (calculated as the average percentage of positive feedback) and RTs throughout the training and experimental sessions for both participant groups across all cues. Specifically, in this context, accuracy and RTs were examined using independent mixed-design repeated measures ANOVAs, which accounted for the primary effects of—and the interactions among—the experimental manipulations (i.e., training session: 2 cues, 2 blocks, and 2 groups; Experimental session: 3 cues, 10 blocks, and 2 groups).

Concerning electrophysiological measures, FRN amplitude, and latency were analyzed using a mixed-design repeated measures ANOVA with the between-subjects factors dominance (high dominance vs. low dominance), and the within-subject factors condition (EXP-certain vs. EXP-uncertain vs. UNEXP-uncertain), valence (positive vs. negative), and electrode (Fz, FCz, and Cz). For P3 amplitude and latency in Pz, a similar analysis was performed, using a mixed-design repeated measures ANOVA with the same between-subjects and within-subject factors. Subsequently, RPE signals (the difference between positive and negative feedback conditions) were further analyzed using a mixed-design repeated measures ANOVA with group (high dominance vs. low dominance), and condition (EXP-certain vs. EXP-uncertain vs. UNEXP-uncertain). In addition, Greenhouse-Geisser correction was used when necessary for adjusting degrees of freedom. The Bonferroni procedure was applied to adjust planned pairwise comparisons for statistically significant interactions. Moreover, to quantify the impact of the experimental manipulation, partial eta-squared ($\eta_p^2$) is reported as the effect size, where a value of 0.05 signifies a small effect, 0.10 indicates a medium effect, and 0.20 represents a large effect [69].

Furthermore, correlational analyses were conducted to explore the possible relationship between personality scores and behavioral data (RT), as well as electrophysiological measures. In this process, both Spearman's rho test and simple linear regression analyses were utilized.

SPSS 26 (SPSS Inc., IBM Corporation, NY), and GraphPad Prism 10.1.0.316 were used for statistical analysis, and alpha was set at $p = 0.05$.

## 3. Result

### 3.1. Behavioral results

#### 3.1.1 Positive feedback, and subjective estimates.
During the training phase, participants properly learned the concept of a cue-response relationship; in the group-based separation, participants with low dominance received positive feedback at an average rate of 86.33% after Cue 1 and 32.90% after Cue 2. For high dominance individuals, the corresponding rates were 83.98% after Cue 1 and 33.27% after Cue 2. Additionally, the subjective estimations of the reward probabilities after training were 86.09% for Cue 1 and 48.04% for Cue 2 for the low-dominance group, whereas the high-dominance group responded with 89.06% for Cue 1 and 42.50% for Cue 2. Pairwise comparisons between the low and high dominance groups revealed no significant differences in any of the examined cues (all $U < 239.5$, all $ps > 0.11$). The Wilcoxon test (because the data did not meet the requirements for parametric testing), was used to analyze the priority of button clicks among participants with high and low dominance. The results showed that participants with low dominance selected button 1 more frequently after Cue 1 than after Cue 2 ($Z = -4.14$; $p < 0.001$) or Cue 3 ($Z = -4.05$; $p < 0.001$). They also chose button 2 more often after Cue 2 than after Cue 1 ($Z = -4.2$; $p < 0.001$) or Cue 3 ($Z = -3.86$; $p < 0.001$). Similarly, high-dominance participants selected button 1 more frequently after Cue 1 than after Cue 2 ($Z = -4.42$; $p < 0.001$) or Cue 3 ($Z = -4.33$; $p < 0.001$), and they selected button 2 more often after Cue 2 than after Cue 1

(Z = −4.51; p < 0.001) or Cue 3 (Z = −3.69; p < 0.001). These findings suggest that participants learned to associate Cue 1 with Button 1 and Cue 2 with Button 2 but did not show a clear preference for button selection after Cue 3.

During the Experiment phase, the low-dominance group achieved positive feedback of 75.80% (Cue 1), 29.39% (Cue 2), and 18.52% (Cue 3). The high-dominance group responded with a similar performance: 76.97% (Cue 1), 33.72% (Cue 2), and 17.98% (Cue 3). Following the experiment, participants estimated the reward probabilities for each cue. The low-dominance group's estimations were 80.00% (Cue 1), 38.70% (Cue 2), and 21.09% (Cue 3). The high-dominance group provided slightly higher estimates: 83.04% (Cue 1), 42.43% (Cue 2), and 19.82% (Cue 3). When comparing the low-and high-dominance groups, there were no significant differences found in any of the examined cues (all U < 252.0, all ps > 0.080). It is concluded that the real frequencies of the three cues were underestimated than subjective estimation. The average positive feedback and subjective estimates in the training and experiment phases for high- and low-dominance are shown in Fig 3.

**3.1.2 RT.** The ANOVA revealed a non-significant main effect for condition [$F_{(1.55, 75.88)}$ = 2.49, p = 0.10, $\eta_p^2$ = 0.048], a non-significant main effect for valence [$F_{(1, 49)}$ = 0.01, p = 0.93, $\eta_p^2$ = 0.000], and for group [$F_{(1, 49)}$ = 0.03, p = 0.87, $\eta_p^2$ = 0.001] (Fig 4). But there was a significant condition × valence interaction [$F_{(1.66, 81.33)}$ = 59.38, p < 0.001, $\eta_p^2$ = 0.548], and condition × group interaction [$F_{(2, 98)}$ = 4.28, p = 0.016, $\eta_p^2$ = 0.080]. No further interaction reached significance (all F < 0.3.47, p > 0.07).

The pairwise comparison results for disentangling the condition × valence interaction showed that when the valence was split, significant differences were found for positive valence. Distinctively, for positive feedback, there was a significant difference between the UNEXP-uncertain and EXP-uncertain (p < 0.001) and EXP-certain (p < 0.001) conditions, with longer RT in UNEXP-uncertain than in the other conditions ($M_{(EXP-certain)}$ = 0.407, $M_{(EXP-uncertain)}$ = 0.521, $M_{(UNEXP-uncertain)}$ = 0.589). For negative valence, there was a significant difference between the UNEXP-uncertain and EXP-uncertain (p < 0.001) and EXP-certain (p < 0.001) conditions, with shorter RTs in UNEXP-uncertain than in the other conditions ($M_{(EXP-certain)}$ = 0.567, $M_{(EXP-uncertain)}$ = 0.525, $M_{(UNEXP-uncertain)}$ = 0.409). The comparison of positive and negative valence for each condition demonstrated significant differences between the EXP-certain positive and EXP-certain negative conditions (p < 0.001), denoting that RTs were longer in negative than positive feedback ($M_{(positive)}$ = 0.407, $M_{(negative)}$ = 0.567). Conversely, there were significant differences between the UNEXP-uncertain positive and UNEXP-uncertain negative conditions (p < 0.001), denoting RTs were shorter in negative than positive feedback ($M_{(positive)}$ = 0.589, $M_{(negative)}$ = 0.409). However, for the EXP-uncertain condition, there was no significant difference between the positive and negative feedback (p = 0.16).

For disentangling the condition × group interaction, separating by group, results indicated that, in the low-dominance group, there were no significant differences in RTs between the EXP-certain vs. EXP-uncertain, and UNEXP-uncertain conditions or between the EXP-uncertain and UNEXP-uncertain conditions (all ps > 0.9). In contrast, for the high-dominance group, the RTs were comparable between the UNEXP-uncertain and EXP-uncertain conditions (p = 0.09), and EXP-certain (p = 0.18), but there were significant differences between the EXP-uncertain vs. the EXP-certain (p = 0.010) conditions. Specifically, the RTs were longer for the EXP-uncertain condition than EXP-certain ($M_{(EXP-certain)}$ = 0.487, $M_{(EXP-uncertain)}$ = 0.560). The results of separating by condition showed that there were non-significant differences in RTs between the low- and high-dominance groups for EXP-certain, EXP-uncertain, and UNEXP-uncertain conditions (all ps > 0.25).

**3.1.3 Learning rate.** One behavioral consequence that can be interpreted through statistical analysis is the learning effect, particularly as reflected in changes in accuracy and RT during the training and experiment phases across blocks. Fig 5 illustrates this trend by showing the average percentage of positive feedback and RT shifts for each cue, separately for the low- and high-dominance groups.

The 60 training trials were divided into two blocks. In the low-dominance group, the mean percentage of positive feedback for Cue 1 increased from 74.55% in Block 1 to 91.96% in Block 2; for Cue 2, it rose from 29.44% to 43.62%. A similar trend was observed in the high-dominance group: Cue 1 increased from 76.85% to 92.65%, and Cue 2 from 25.72% to 40.19% (see Fig 5a).

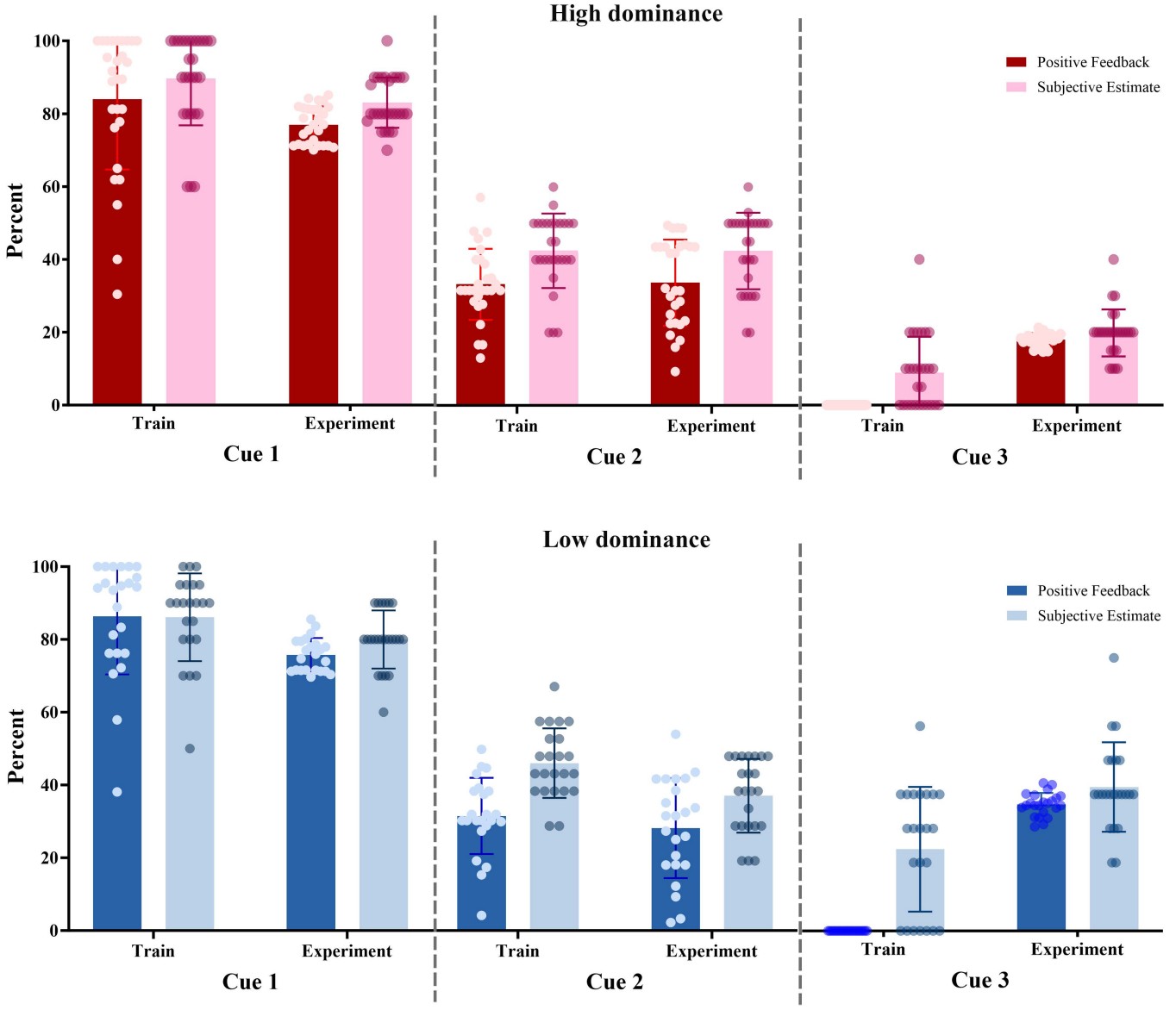

**Fig 3. Average positive feedback and average subjective estimates for positive feedback for each cue for the train and experiment session; in percentage.** Error bars denote SD.

Simple linear regression analyses confirmed statistically significant learning effects in all cues in both groups, with positive slopes indicating systematic improvements (high-dominance_cue1: 15.80 (p = 0.006), cue2: 14.47 (p < 0.001); low-dominance_cue1: 17.41 (p = 0.007), and cue2: 14.19 (p < 0.001)), each with modest but meaningful $R^2$ values (ranging from 0.13 to 0.29). Therefore, accuracy improved reliably with practice. Mixed repeated-measures ANOVA further supported these findings, revealing significant main effects for cue [$F_{(1, 49)}$ = 252.39, p < 0.001, $\eta_p^2$ = 0.837] and block [$F_{(1, 49)}$ = 97.82, p < 0.001, $\eta_p^2$ = 0.666], but not for group [$F_{(1, 49)}$ = 0.14, p = 0.707, $\eta_p^2$ = 0.003].

Paired samples t-tests revealed significant improvements across blocks for both groups. In the low-dominance group, Cue 1 rose from 74.55% to 91.96% and Cue 2 from 29.44% to 43.63% (both p < 0.001). Similarly, in the high-dominance

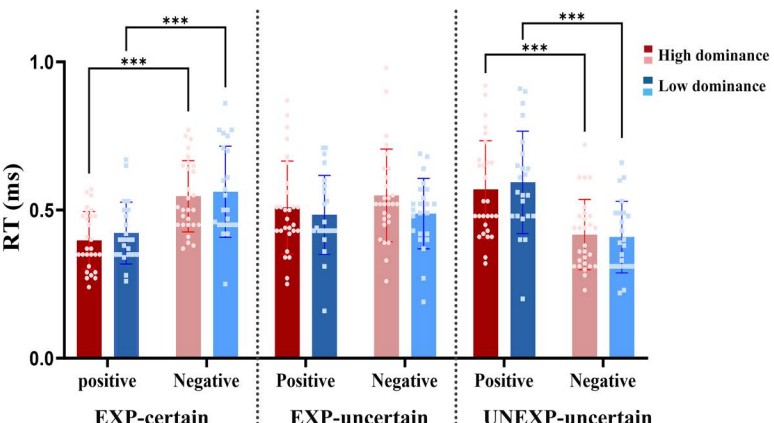

**Fig 4. Mean of RT (ms) for all conditions in high and low dominance groups.** Error bars denote SD. In all comparisons, "***" denotes p<0.001.

group, Cue 1 improved from 76.85% to 92.65%, and Cue 2 from 25.72% to 40.19% (both p<0.001). These findings suggest that both groups demonstrated comparable learning trajectories. Notably, the low-dominance group showed a slightly larger improvement for Cue 1 (17.41 vs. 15.80), although they started from a marginally lower baseline. For Cue 2, both groups improved by nearly the same magnitude, though the low-dominance group began with a higher initial accuracy. Moreover, the high-dominance group displayed less variability in Block 2, indicating more consistent performance.

In contrast, during the experiment session, no such learning trend emerged. For the low-dominance group, mean accuracy for Cue 1 increased slightly from 68.66% to 73.72%, while Cue 2 decreased marginally from 29.66% to 29.02%, and Cue 3 remained virtually unchanged (15.45% to 15.66%). The high-dominance group showed similar trends: Cue 1 increased from 70.07% to 74.68%, while Cue 2 and Cue 3 declined slightly (31.76% to 30.33% and 15.38% to 13.75%, respectively) (Fig 5b). Linear regression analyses yielded non-significant slopes for all six conditions, indicating no meaningful learning effect across the blocks. Slopes ranged from –0.0884 to 0.3355, with all p>0.1, and $R^2$ values near zero, suggesting stable performance without systematic improvement.

A mixed repeated-measures ANOVA during the experimental session confirmed a significant main effect for cue [$F_{(1.24, 60.86)}$ = 1492.13, p<0.001, $\eta_p^2$ = 0.968], with higher accuracy for Cue 1 than Cues 2 and 3, and a significant main effect for block [$F_{(5.71, 279.86)}$ = 3.3, p=0.004, $\eta_p^2$ = 0.063]. However, no significant main effect for group was found [$F_{(1, 49)}$ = 1.20, p=0.279, $\eta_p^2$ = 0.024]. A significant interaction emerged between cue × block [$F_{(6.48, 317.59)}$ = 2.77, p=0.010, $\eta_p^2$ = 0.054], while no other interactions reached significance.

Post-hoc analyses of the cue × block interaction revealed selective effects. For Cue 1, several blocks showed significant differences, suggesting minor fluctuations in performance. For example, Block 1 differed significantly from Block 5 (p=0.049), and Block 3 from Block 8 (p=0.002). Cue 2 exhibited only one significant difference (Block 2 vs. Block 5, p=0.040), while Cue 3 showed two (Block 2 vs. Block 10, and Block 5 vs. Block 10, p<0.001 and p=0.006, respectively), indicating minimal variation over time.

In terms of RTs during the training session, both dominance groups showed general improvements. In the low-dominance group, RTs for Cue 1 declined from 0.73 to 0.65 seconds and for Cue 2 from 0.75 to 0.59. The high-dominance group displayed similar trends, with Cue 1 decreasing from 0.69 to 0.58 seconds and Cue 2 from 0.76 to 0.63 (Fig 5c). Regression analysis revealed negative slopes for both cue 1 and cue 2 in both groups but only cue 2 in low-dominance group was statistically significant ($R^2$=0.1403, $F_{(1,44)}$ = 7.181, p=0.010), suggesting that while all conditions trended downward, only one showed a robust practice-related RT improvement.

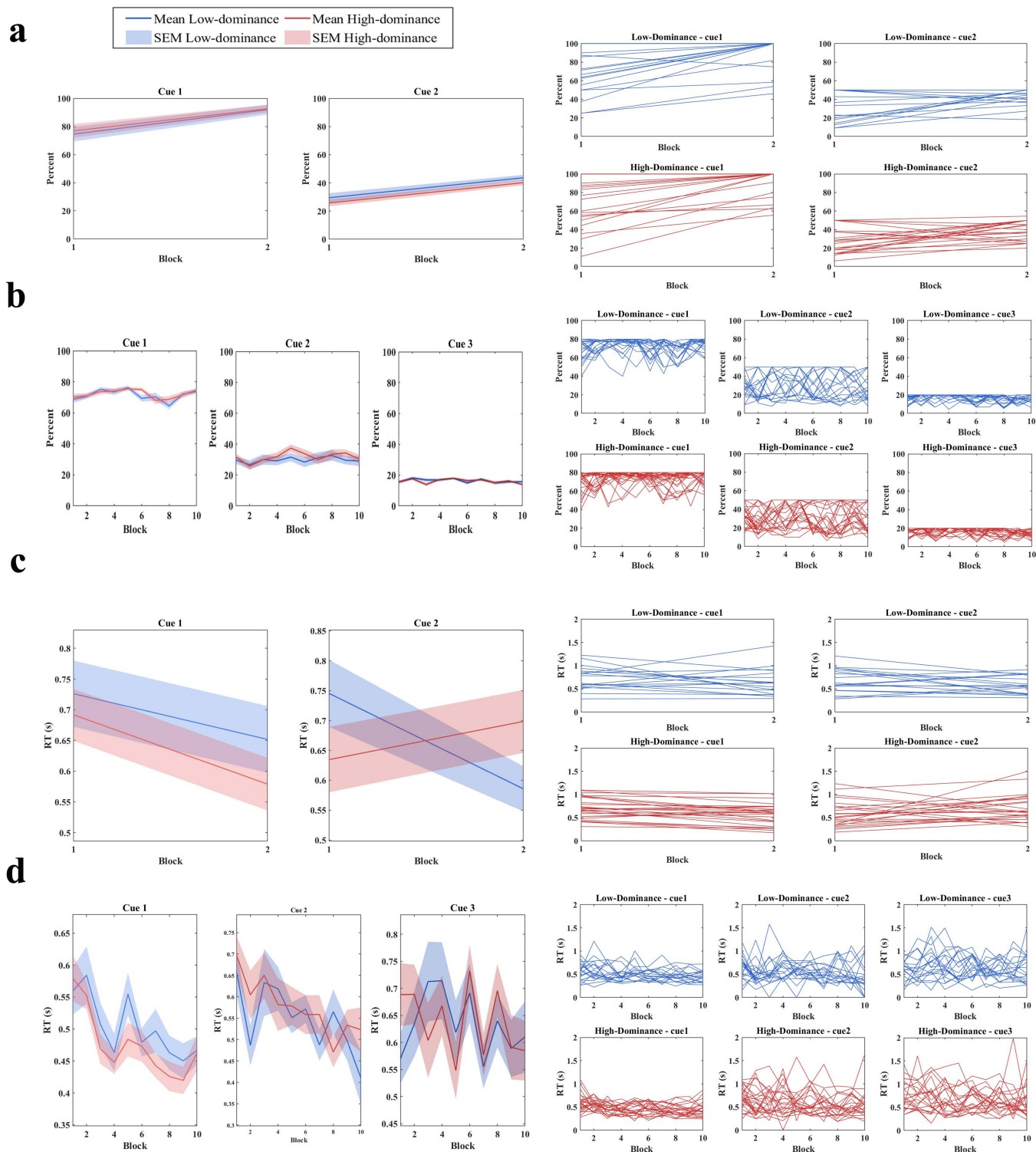

**Fig 5. The learning effect during the training and experiment phases across blocks, particularly as reflected in changes in accuracy (percentage of positive feedback) and RT;** (a) left side: average of accuracy curves for cues in train session, right side: individual accuracy curves per cue and group in train session; (b) average of accuracy curves for cues in experiment session, right side: individual accuracy curves per cue and group in experiment session; (c) left side: RT in train session, right side: individual RT curves per cue and group in train session; (d) left side: RT in experiment session, right side: individual RT curves per cue and group in experiment session. Thick lines characterize the mean across task block and shaded areas indicate the standard error of the mean (SEM).

Mixed repeated-measures ANOVA on training RTs revealed a significant main effect of block [$F_{(1, 49)}$ = 22.87, $p < 0.001$, $\eta_p^2$= 0.318], indicating faster responses in the second block. There were no significant main effects for cue [$F_{(1, 49)}$ = 0.23, $p = 0.630$, $\eta_p^2$ = 0.005]; and group [$F_{(1,49)}$ = 0.08, $p = 0.775$, $\eta_p^2$ = 0.002]. Paired sample tests confirmed significant RT reductions in the high-dominance group for both cues (Cue 1: $p = 0.002$; Cue 2: $p = 0.008$). In contrast, only Cue 2 reached significance in the low-dominance group ($p = 0.008$), with Cue 1's reduction falling short ($p = 0.253$). This suggests more consistent and generalized learning effects in the high-dominance group. RT variability was also higher in the low-dominance group, particularly for Cue 1, potentially accounting for the weaker effect.

During the experimental session, RTs also showed overall reductions, indicating a speed-accuracy trade-off. In the low-dominance group, Cue 1 decreased from 0.56 to 0.47 seconds and Cue 2 from 0.66 to 0.41, while Cue 3 slightly increased. In the high-dominance group, Cue 1 dropped from 0.58 to 0.46, Cue 2 from 0.70 to 0.52, and Cue 3 from 0.69 to 0.60 (Fig 5d).

Simple linear regressions for RT during the experiment revealed significant negative slopes in cue 1, and cue 2 in both groups, (Slopes ranged from –0.01192 to –0.01853, all $p < 0.0025$), again indicating modest but consistent learning effects. While cue 3 in low/high-dominance group remained non-significant (all P value > 0.21). Despite low R² values (low-dominance: –0.0056, high-dominance: –0.0020), these patterns confirm that repeated practice contributed to slight but reliable reductions in RT.

A mixed repeated-measures ANOVA of RT during the experiment confirmed a significant main effect for cue [$F_{(1.74, 85.63)}$ = 32.24, $p < 0.001$, $\eta_p^2$= 0.397], with Cue 1 yielding the fastest responses. The main effect for block was also significant [$F(6.31, 309.52)$ = 5.97, $p < 0.001$, $\eta_p^2$= 0.109], highlighting that responses were generally quicker by Block 10. No significant effect for group was found. Additionally, a significant interaction between cue and block emerged [$F_{(8.25, 404.71)}$ = 2.61, $p = 0.008$, $\eta_p^2$= 0.051], with post-hoc comparisons confirming that Cue 1 showed the clearest pattern of decline over time (Block 1 vs. Block 4: $p = 0.005$; Block 1 vs. Block 8: $p = 0.006$; Block 1 vs. Block 9: $p = 0.004$; Block 2 vs. Block 4: $p = 0.003$; Block 2 vs. Block 6: $p = 0.020$; Block 2 vs. Block 7: $p = 0.002$; Block 2 vs. Block 8: $p = 0.001$; Block 2 vs. Block 9: $p < 0.001$; Block 2 vs. Block 10: $p = 0.006$) particularly between early and late blocks. Cue 2 displayed a similar, though less consistent (Block 1 vs. Block 7: $p = 0.008$; Block 1 vs. Block 8: $p = 0.026$; Block 1 vs. Block 9: $p = 0.005$; Block 1 vs. Block 10: $p = 0.014$), trend. Cue 3 remained largely stable across groups and blocks, with minimal changes.

## 3.2 ERP data

For electrode locations Fz, FCz, and Cz, where the FRN was analyzed, and for Pz, where the P3 ERPs were recorded under the three conditions (EXP-certain, EXP-uncertain, and UNEXP-uncertain) for both positive and negative valences, as well as RPE signals (negative minus positive conditions; Picton et al., 2000 [70]). Figs 6 and 7 illustrate the amplitude waveforms and scalp topographies of FRN (at the Fz electrode, known for its prominent FRN amplitudes) and P3 components.

### 3.2.1 FRN.  3.2.1.1. Amplitude

The ANOVA of the FRN amplitude revealed a significant main effect for electrode [$F_{(2,98)}$ = 7.08, $p = 0.001$, $\eta_p^2$ = 0.126], with more negative FRN amplitude for Fz than FCz and Cz electrodes ($M_{(Fz)}$= 2.025, $M_{(FCz)}$= 2.929, $M_{(Cz)}$= 2.980) (each electrode was analyzed independently, see supplementary material for more details, S2, S3, and S4 Files), a non-significant main effect for condition, [$F_{(1.68, 82.2)}$ = 0.17, $p = 0.80$, $\eta_p^2$ = 0.003], a significant main effect for valence [$F_{(1, 49)}$ = 10.55, $p = 0.002$, $\eta_p^2$= 0.177], with larger FRN amplitude for negative than positive feedback ($M_{(positive)}$= 3.856, $M_{(negative)}$= 1.433), and for group [$F_{(1,49)}$ = 14.20, $p < 0.001$, $\eta_p^2$ = 0.225], with larger FRN amplitude in the low-dominance than the high-dominance group ($M_{(high-dominance)}$= 6.046, $M_{(low-dominance)}$= −0.757, Fig 6). In addition, there was a significant interaction between condition × valence [$F_{(2, 98)}$ = 7.00, $p = 0.001$, $\eta_p^2$ = 0.125]. No further interaction reached significance (all F < 2.1, $p > 0.13$).

Separating by valence, the post-hoc analysis of the condition × valence interaction revealed non-significant condition difference was observed for positive EXP-certain vs. EXP-uncertain ($p = 0.344$) and UNEXP-uncertain ($p = 0.075$)

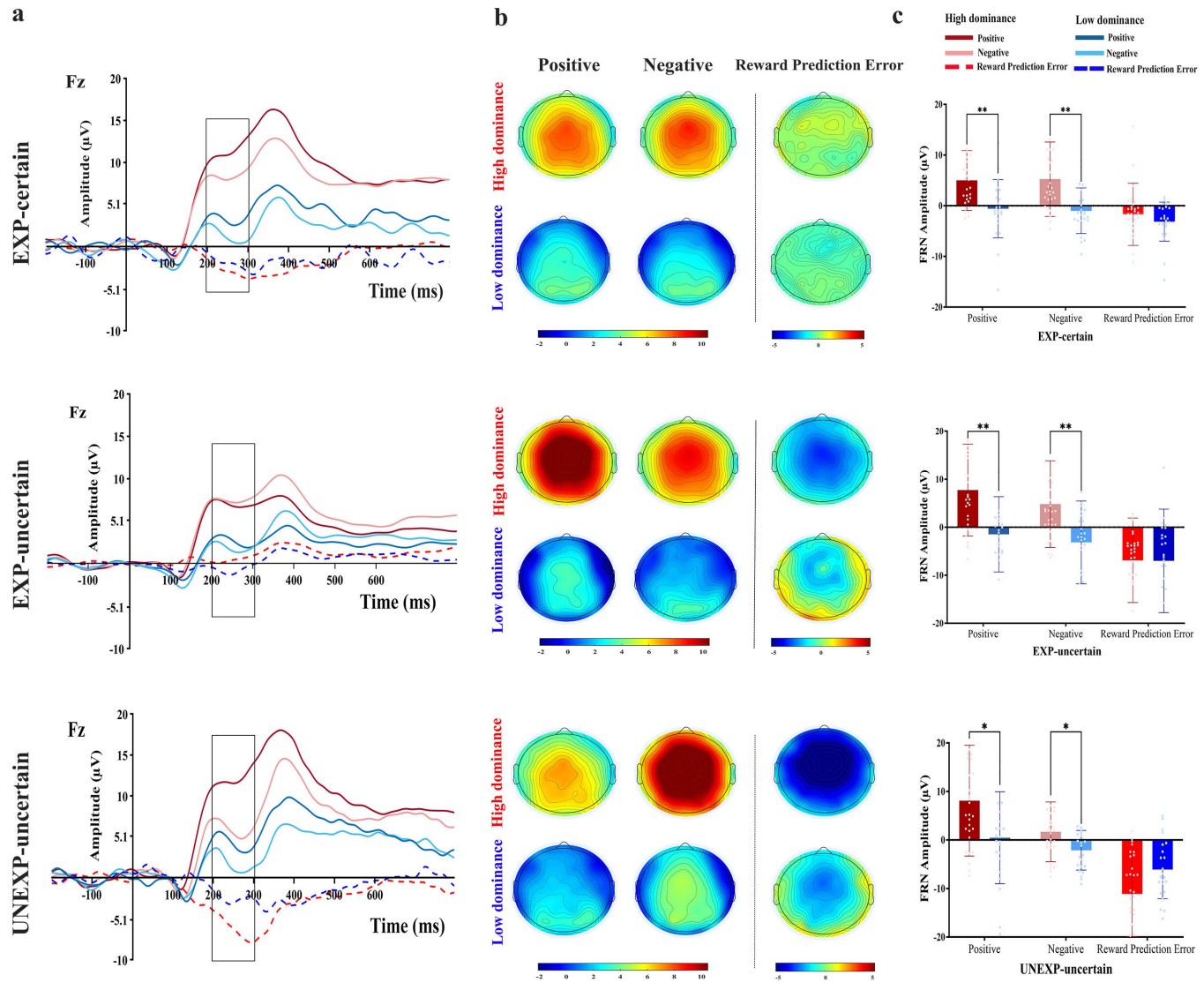

**Fig 6. FRN component.** a) Grand averaged ERP waveforms at Fz electrode for low and high dominance groups in positive, negative feedback, and RPE (difference waves amplitude courses; negative minus positive) in three conditions (EXP-certain, EXP-uncertain, and UNEXP-uncertain). b) Topographical scalp for the difference waves FRN component (200–300 ms post-feedback window) for each of the positive, negative, and RPE conditions for high and low dominance groups; in μV. C) FRN means amplitude differences; Error bars denote SD. In all comparisons, "**" denotes p < 0.01, and "*" indicates p < .05 in all comparisons.

conditions, as well as in the EXP-uncertain vs. UNEXP-uncertain (p = 0.86) conditions. For negative valence, there was a significant difference between the EXP-certain vs. UNEXP-uncertain conditions (p < 0.001), with a larger FRN amplitude observed in the UNEXP-uncertain condition than in the EXP-certain condition ($M_{(EXP-certain)}$ = 2.737, $M_{(UNEXP-uncertain)}$ = −0.042). The conditions of the EXP-uncertain vs. EXP-certain (p = 0.98) and UNEXP-uncertain (p = 0.47) did not show significant differences. When separated by condition, there was a comparable valence effect for the EXP-certain condition (p = 0.90); but there was a marginally significant valence effect for the EXP-uncertain ($M_{(positive)}$ = 4.029, $M_{(negative)}$ = 1.604, p = 0.051) and a significant UNEXP-uncertain ($M_{(positive)}$ = 4.911, $M_{(negative)}$ = −0.042, p < 0.001) conditions, with a larger FRN amplitude observed for negative than positive feedback.

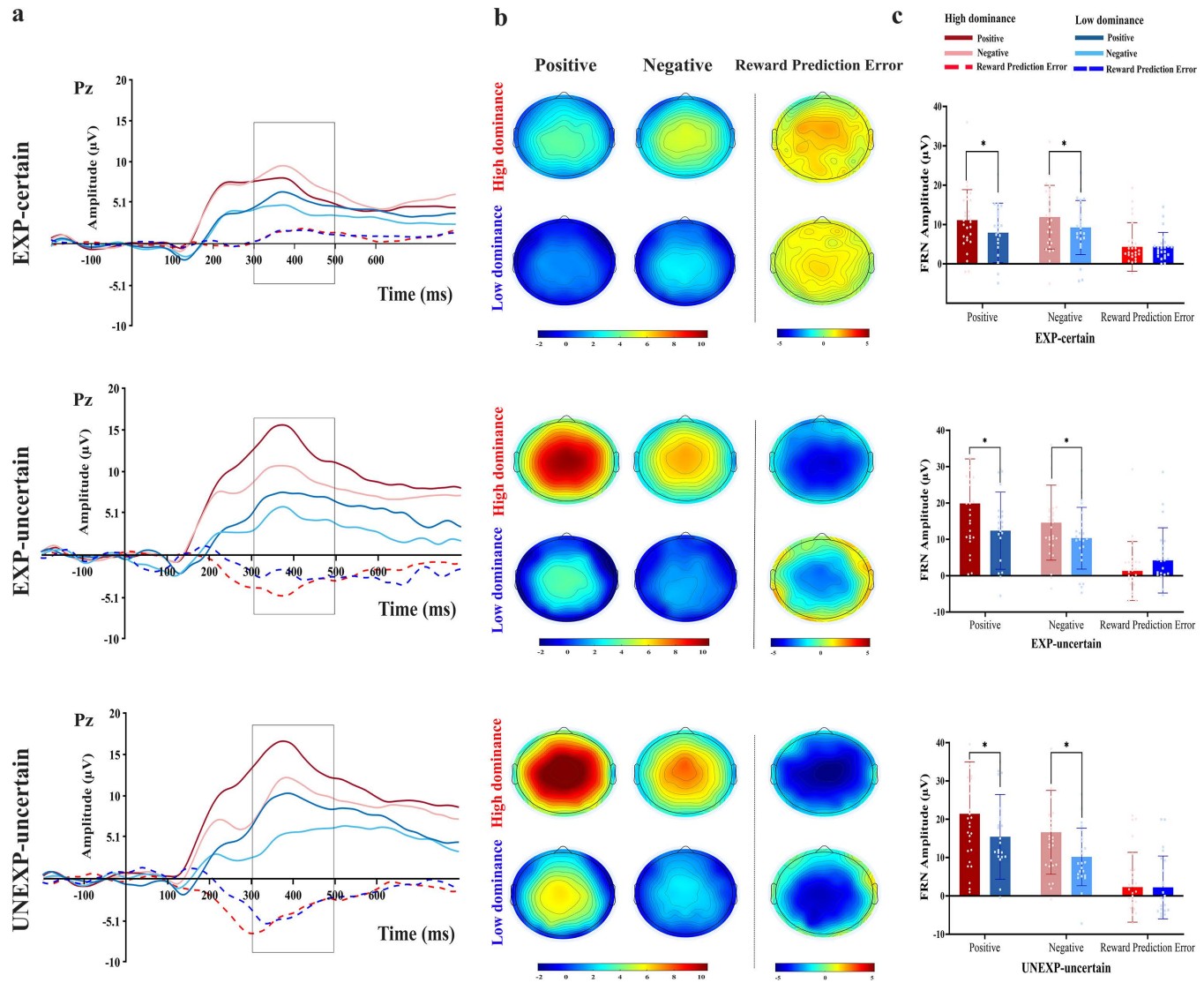

**Fig 7. P3 component.** a) Grand averaged ERP waveforms at Pz for low and high dominance groups in positive, negative feedback, and RPE (difference waves amplitude courses; negative minus positive) in three conditions (EXP-certain, EXP-uncertain, and UNEXP-uncertain). b) Topographical scalp for the difference waves P3 component (300–500 ms post-feedback window) for each of the positive, negative, and RPEs for high and low dominance groups; in μV. C) P3 means amplitude differences; Error bars denote SD. In all comparisons "*" indicates p < .05 in all comparisons.

### 3.2.1.2. RPE

The grand mean difference wave amplitudes at the Fz electrode were used to assess RPE signal in the FRN component as the largest FRN amplitudes were shown at Fz compared to FCz and Cz in our ANOVA (Fz vs. FCz: p = 0.012, Fz vs. Cz: p = 0.005; see 3.2.1.1. Amplitude).

The ANOVA of the RPE indicated a significant main effect for condition [$F_{(2, 98)}$ = 10.53, p < 0.001, $\eta_p^2$ = 0.177], with larger RPE for UNEXP-uncertain than EXP-uncertain and EXP-certain condition ($M_{(EXP-certain)}$ = −2.424, $M_{(EXP-uncertain)}$ = −6.952, $M_{(UNEXP-uncertain)}$ = −8.532), non-significant main effect for group [$F_{(1, 49)}$ = 0.65, p = 0.42, $\eta_p^2$ = 0.013] (Fig 6). However, there was a significant interaction between condition × group [$F_{(2, 98)}$ = 3.19, p = 0.046, $\eta_p^2$ = 0.061].

To disentangle the condition × group interaction, when analysis separated by group, the results indicated that in the low-dominance group, there were no significant differences in all conditions (all Ps > 0.27). For the high-dominance group, there was a comparable RPE between EXP-uncertain and UNEXP-uncertain conditions (p = 0.07), but there were significant differences between the EXP-certain vs. the EXP-uncertain (p = 0.042), or UNEXP-uncertain (p < 0.001) conditions. Specifically, the RPE signal was more negative for the UNEXP-uncertain condition than other conditions ($M_{(EXP-certain)}$ = −1.709, $M_{(EXP-uncertain)}$ = −6.895, $M_{(UNEXP-uncertain)}$ = −11.112). Separating the analysis by condition, a more negative RPE amplitude was found in high-dominance than low-dominance for the UNEXP-uncertain condition ($M_{(high-dominance)}$ = −11.112, $M_{(low-dominance)}$ = −5.592, p = 0.020). However, no significant differences were observed between the low- and high-dominance groups for the EXP-certain (p = 0.34) and EXP-uncertain (p = 0.97) conditions.

### 3.2.1.3. Latency

The ANOVA of the FRN latency indicated a non-significant main effect for electrode [$F_{(1.49, 72.97)}$ = 1.86, p = 0.17, $\eta_p^2$ = 0.037], a non-significant main effect for condition [$F_{(2,98)}$ = 1.56, p = 0.22, $\eta_p^2$ = 0.031], a significant main effect for valence [$F_{(1, 49)}$ = 4.69, p = 0.035, $\eta_p^2$ = 0.087], with shorter FRN latency for negative than positive feedback ($M_{(positive)}$ = 259.974, $M_{(negative)}$ = 254.586), and a non-significant main effect for group [$F_{(1, 49)}$ = 1.73, p = 0.19, $\eta_p^2$ = 0.034]. In addition, there was a significant valence × group interaction [$F_{(1,49)}$ = 8.82, p = 0.005, $\eta_p^2$ = 0.153]. No further interaction reached significance (all F < 2.31, p > 0.11).

After separating the participants into low- and high-dominance groups, the post-hoc analysis of the valence × group interaction revealed that for the low-dominance group, latency for negative feedback was significantly shorter than for positive feedback ($M_{(positive)}$ = 267.003, $M_{(negative)}$ = 254.227, p = 0.001), while this effect was not observed in the high-dominance group (p = 0.55). Direct comparing the two groups in terms of positive or negative valence indicated a significant difference between the low- and high-dominance groups (p = 0.019) for positive valence, where the low-dominance group exhibited longer FRN latency than the high-dominance group ($M_{(high\ dominance)}$ = 252.945, $M_{(low\ dominance)}$ = 267.003), while no significant effect was observed for negative valence (p = 0.90).

### 3.2.2 P3. 3.2.2.1. Amplitude

The ANOVA of the P3 amplitude showed a significant main effect for condition [$F_{(2, 98)}$ = 28.98, p < 0.001, $\eta_p^2$ = 0.372], with larger P3 amplitude for UNEXP-uncertain than EXP-uncertain and EXP-certain condition ($M_{(EXP-certain)}$ = 10.013, $M_{(EXP-uncertain)}$ = 14.208, $M_{(UNEXP-uncertain)}$ = 15.892), for valence [$F_{(1, 49)}$ = 12.72, p = 0.001, $\eta_p^2$ = 0.206], with larger P3 amplitude in positive than negative feedback ($M_{(positive)}$ = 14.622, $M_{(negative)}$ = 12.121), and for group [$F_{(1, 49)}$ = 4.22, p = 0.045, $\eta_p^2$ = 0.079], with larger P3 amplitude for high-dominance group than low-dominance group ($M_{(high\ dominance)}$ = 15.851, $M_{(low\ dominance)}$ = 10.891, Fig 7). Also, there was a significant condition × valence interaction [$F_{(2,98)}$ = 9.7, p < 0.001, $\eta_p^2$ = 0.165]. No further interaction reached significance (all F < 2.52, p > 0.085).

Dividing the analysis by valence, the post-hoc results of the condition × valence interaction revealed a significant difference between EXP-certain positive and both other positive conditions (EXP-uncertain: p < 0.001, UNEXP-uncertain: p < 0.001), with a larger P3 amplitude in UNEXP-uncertain and EXP-uncertain conditions ($M_{(EXP-certain)}$ = 9.486, $M_{(EXP-uncertain)}$ = 15.972, $M_{(UNEXP-uncertain)}$ = 18.407). However, there was no significant difference in P3 amplitude between EXP-uncertain and UNEXP-uncertain (p = 0.11). For negative valence, UNEXP-uncertain showed a larger P3 amplitude than EXP-certain ($M_{(EXP-certain)}$ = 10.539, $M_{(UNEXP-uncertain)}$ = 13.378, p = 0.014), while the other conditions did not differ significantly (UNEXP-uncertain vs. EXP-uncertain: p = 0.93; EXP-uncertain vs. EXP-certain: p = 0.13). When separated by condition, the P3 amplitude was larger after EXP-uncertain positive compared to negative ($M_{(positive)}$ = 15.972, $M_{(negative)}$ = 12.445, p = 0.006), and for UNEXP-uncertain positive compared to negative ($M_{(positive)}$ = 18.407, $M_{(negative)}$ = 13.378, p < 0.001), while no significant difference emerged for EXP-certain positive compared to negative (p = 0.19).

### 3.2.2.2. RPE

The grand mean difference wave amplitudes method in the Pz electrode was used to evaluate neural activity for RPEs in P3 components.

The ANOVA of the RPE indicated non-significant main effect for condition [$F_{(2, 98)}$ = 1.34, p = 0.27, $\eta_p^2$ = 0.027], for group [$F_{(1, 49)}$ = 0.45, p = 0.51, $\eta_p^2$ = 0.009] (Fig 7), and for condition × group interaction [$F_{(2, 98)}$ = 0.78, p = 0.46, $\eta_p^2$ = 0.016].

### 3.2.2.3. Latency

The ANOVA of the P3 latency demonstrated a significant main effect for condition [$F_{(2, 98)}$ = 4.27, p = 0.017, $\eta_p^2$ = 0.080], with longer P3 latency for UNEXP-uncertain than EXP-uncertain and EXP-certain condition ($M_{(EXP-certain)}$ = 385.941, $M_{(EXP-uncertain)}$ = 389.445, $M_{(UNEXP-uncertain)}$ = 403.164), for valence [$F_{(1,49)}$ = 4.98, p = 0.030, $\eta_p^2$ = 0.092], with longer P3 latency for negative than positive feedback ($M_{(positive)}$ = 387.632, $M_{(negative)}$ = 398.069), and a non-significant main effect for group [$F_{(1,49)}$ = 3.68, p = 0.061, $\eta_p^2$ = 0.070]. Also, there was a significant condition × valence interaction [$F_{(2, 98)}$ = 3.9, p = 0.024, $\eta_p^2$ = 0.074]. No further interaction reached significance (all F < 0.16, p > 0.74).

To decipher the significant condition × valence interaction, post-hoc tests revealed that for positive valence, there were no significant differences between the EXP-certain vs. EXP-uncertain, UNEXP-uncertain, and EXP-uncertain vs. UNEXP-uncertain conditions (all ps > 0.9) when separated by valence. For negative valence, there was a significant difference between the UNEXP-uncertain vs. EXP-uncertain (p = 0.001), and EXP-certain (p = 0.007) conditions, with longer P3 latency observed in the UNEXP-uncertain condition than in the other two conditions ($M_{(EXP-certain)}$ = 386.886, $M_{(EXP-uncertain)}$ = 390.164, $M_{(UNEXP-uncertain)}$ = 417.156). Separating by condition, a comparable valence effect was observed for the EXP-certain (p = 0.80) and EXP-uncertain (p = 0.85) conditions, while a significant valence effect was observed for the UNEXP-uncertain condition (p = 0.002), with a longer P3 latency observed for negative feedback compared to positive feedback ($M_{(positive)}$ = 389.172, $M_{(negative)}$ = 417.156).

### 3.3 Correlational analyses

Spearman's rho correlation and simple linear regression analyses were conducted to examine the predictive relationship between PRF-d scores and RT, FRN, P3 amplitudes, and their corresponding RPE signals. Results indicated no significant correlation between PRF-d scores and RT across any condition (all $R^2$ < 0.02, p > 0.32) (Fig 8a).

However, a significant relationship emerged between PRF-d scores and the mean amplitude of FRN in several experimental conditions. Specifically, PRF-d scores significantly predicted FRN amplitude in the EXP-certain positive ($R^2$ = 0.11, $F_{(1, 49)}$ = 6.33, p = 0.015), EXP-certain negative ($R^2$ = 0.13, $F_{(1, 49)}$ = 7.51, p = 0.009), EXP-uncertain positive ($R^2$ = 0.14, $F_{(1, 49)}$ = 7.82, p = 0.007), EXP-uncertain negative ($R^2$ = 0.14, $F_{(1, 49)}$ = 8.22, p = 0.006), and UNEXP-uncertain negative ($R^2$ = 0.08, $F_{(1, 49)}$ = 4.57, p = 0.038) conditions. A similar trend was observed in the UNEXP-uncertain positive condition ($R^2$ = 0.06, $F_{(1, 49)}$ = 3.36, p = 0.073), although it did not reach statistical significance (Fig 8b). These findings suggest that lower PRF-d scores are associated with more negative FRN amplitudes, reflecting heightened sensitivity to feedback.

A significant positive relationship was also found between PRF-d scores and P3 amplitude in the UNEXP-uncertain negative condition ($R^2$ = 0.09, $F_{(1, 49)}$ = 4.98, p = 0.030). No significant associations were observed between PRF-d scores and P3 amplitude in the remaining conditions (p > 0.07) (Fig 8c).

Regarding the RPE signal derived from FRN, regression analyses revealed a significant relationship between PRF-d scores and RPE in the UNEXP-uncertain condition ($R^2$ = 0.09, $F_{(1, 49)}$ = 4.98, p = 0.030). Specifically, individuals with higher PRF-d scores (i.e., the high-dominance group) exhibited more negative RPEs in this condition. No significant associations were found between PRF-d scores and FRN-based RPEs in other conditions (p > 0.5) (Fig 8d). Likewise, no significant correlations were observed between PRF-d scores and P3-based RPEs across any condition (p > 0.4) (Fig 8e).

## 4. Discussion

This investigation sought to disentangle the intricate interplay between social dominance, neural circuitry, and behavioral tendencies during decision-making under conditions of uncertainty. We aimed to understand how these neural indicators were affected when individuals received positive and negative feedback under different conditions of expectation and uncertainty. To achieve this, we divided participants into high- and low-dominance groups. We subjected them to varying

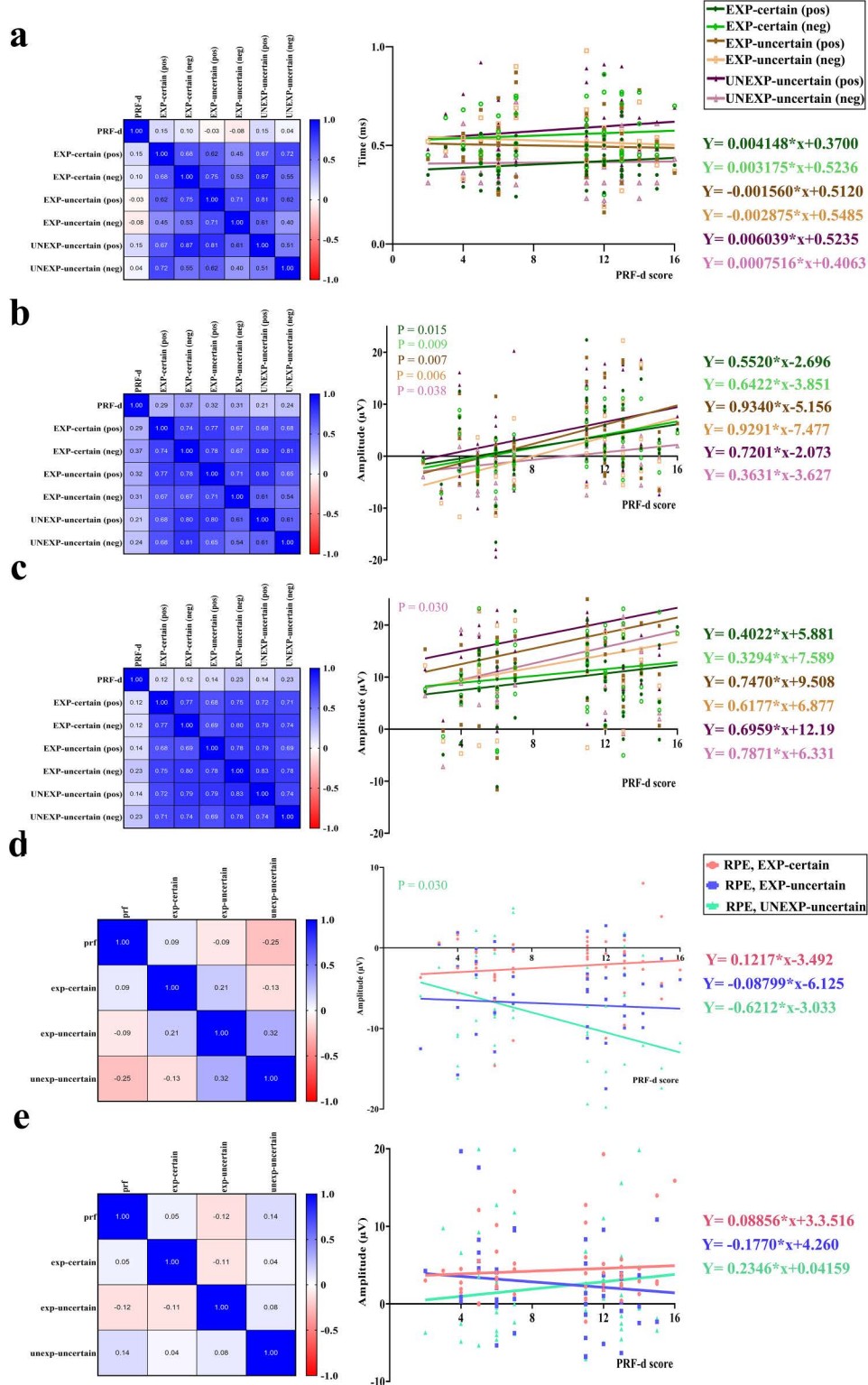

**Fig 8. Correlation matrix (left), and Simple linear regressions (right).** a) PRF-d score and RT; b) PRF-d score and FRN amplitude; c) PRF-d score and P3 amplitude; d) PRF-d score and RPEs of FRN amplitude; e) PRF-d score and RPEs of P3 amplitude.

levels of feedback uncertainty, namely, expected-certain, expected-uncertain, and unexpected-uncertain feedback scenarios (For detailed information on the decision-making paradigm and terminology, please refer to Section 2.2.2. Decision-making paradigm).

During a training session, the participants' expectations were established, and then they were altered during the experimental session. The participants' subjective evaluations and button-pressing behavior showed that they had correctly acquired knowledge of the connections between actions and outcomes.

The amplitude of the FRN and P300 reflects the strength of neural responses to feedback: FRN amplitude is typically more negative following unexpected or negative outcomes and is linked to performance monitoring [7,15,71], while P300 amplitude is more positive for salient or motivationally relevant stimuli, reflecting attentional allocation [13,72]. Latency, defined as the time from feedback onset to ERP peak, indexes the speed of cognitive evaluation, with shorter latencies indicating faster processing. The RPE, operationalized as the difference in amplitude between negative and positive feedback, reflects the neural coding of outcome discrepancies based on expectations, as conceptualized in RL models [23]. Our results suggest that the independent variables (e.g., cue type, dominance group) differentially affected these measures—modulating FRN and P300 amplitudes in line with feedback salience, influencing latency as an index of processing efficiency, and shaping RPE as a marker of prediction updating (e.g., [71,72]).

Our study's results revealed no significant difference in RT between the low and high dominant groups. In keeping with earlier investigations, it was observed that low dominance was associated with larger amplitudes of the FRN compared to high levels of dominance. However, additional factors such as expectation level, uncertainty, and negative valence were also identified as valuable predictors of FRN amplitude. Specifically, it was found that the FRN amplitude was greater in response to unexpected-uncertain conditions, particularly when negative feedback was presented. As well as the P3 amplitude was larger for the high-dominance than for the low-dominance group. Especially, it was larger after the unexpected-uncertain condition, particularly in response to positive feedback. Moreover, PRF-d scores significantly predicted FRN amplitudes across different expectancy and certainty conditions, particular in unexpected uncertain negative condition, with lower dominance scores linked to more negative FRNs. as well as with P3 amplitude in the UNEXP-uncertain negative condition, with higher dominance scores linked to more positive P3 amplitude. Additionally, PRF-d scores showed a positive relationship with FRN-derived RPE in the unexpected uncertain negative condition.

## 4.1. Social dominance, and RT

The results indicated that certain conditions and feedback types could affect RT differently. In detail, participants in the expected-certain positive condition and the unexpected-uncertain negative condition exhibited faster RT. Regarding the effect of conditions and feedback types on RT, it is possible that the RT varied due to the engagement of different cognitive processes in each condition and feedback type. Participants in the expected-certain condition with positive feedback may have been more motivated to perform well and had a greater sense of confidence due to the certainty of the feedback. This idea is supported by a study by Kutas and Donchin (1980) [73], which found that participants exhibited a stronger preparation potential when the response was more certain and interpreted this finding as evidence of increased motivation and confidence in more certain conditions, which could lead to faster and more accurate responses. Moreover, according to Grupe and Nitschke (2013) [74], uncertainty, and anxiety could increase cognitive and physiological arousal, potentially improving performance in certain situations. For instance, in unexpected-uncertain conditions with negative feedback, participants may have had a faster response time due to increased uncertainty and anxiety causing a stronger stress response.

Apart from conditions and valence, our findings revealed an association between social dominance and RT differences. Exclusively, the high-dominance group responded faster to the expected-certain condition compared to both the expected-uncertain and unexpected-uncertain conditions. This observation aligned with established research demonstrating that predictable situations necessitate lower cognitive load due to diminished processing demands placed on the

brain [75,76]. When outcomes were anticipated, the brain optimized processing efficiency, potentially freeing up cognitive resources for faster response execution [77]. This reduced cognitive load in the expected-certain condition for the high-dominance group might be further accentuated by their documented lower mind-reading motivation [78]. This potentially translated to less effort invested in processing unexpected cues, ultimately impacting their response times. Additionally, social dominance was frequently associated with a strong desire for control [79,80]. In the expected-certain condition, participants possess the greatest degree of control as the outcome was anticipated with heightened certainty. This enhanced sense of control could motivate the high-dominance group to respond faster in an attempt to exert greater influence over the situation [44]. In essence, the confluence of reduced cognitive load stemming from predictability, potentially lower effort allocated to processing unexpected cues due to reduced mind-reading motivation, and a heightened desire for control might collectively explain the observed faster RTs exhibited by the high-dominance group in the expected-certain condition.

## 4.2. Social dominance and feedback processing

### 4.2.1 FRN.
The current study's findings indicate that individuals with lower dominance levels exhibited a more pronounced FRN compared to those with higher dominance. This supported previous research which indicated that individuals with low social dominance were more likely to perceive feedback as a threat and experience social evaluation threats [45,46]. The available evidence suggested that individuals with high social status tended to rely on a proactive dorsal control system, while those with low social status were more dependent on a reactive ventral system [46]. The reactive ventral system allowed low-status individuals to monitor and adjust their performance based on external events, but it may have also hindered their ability to filter out distractions. Conversely, individuals with high status tended to adhere to predetermined plans, whereas those with low status continuously monitored their surroundings, looking out for unexpected changes that could indicate potential social exclusion [46]. Additionally, low-dominance individuals may have exhibited a larger FRN response to negative feedback due to their inhibitory behavior and sensitivity to environmental changes [45]. Moreover, lower social-status individuals tended to self-evaluate their performance more than higher social-status individuals [24] and may have also been more likely to experience feelings of shame or guilt in response to negative feedback, which could lead to even larger FRN responses (for further explanation, see [81]).

Furthermore, the FRN was amplified in response to unexpected negative feedback, suggesting heightened sensitivity to negative outcomes in uncertain situations. This finding aligns with previous research demonstrating that expectation and uncertainty levels are significant predictors of FRN amplitude, which is modulated by expectation violations [12,16,22]. RPE findings showed that the mean difference amplitude was the largest after the UNEXP-uncertain condition than EXP-uncertain and EXP-certain conditions. An increase in FRN amplitude was often interpreted as a prominent PE, reflecting the difference between observed outcomes and predicted expectations [12,20,22,82]. In situations where individuals are confronted with uncertain results, their intensified neural reaction could signify heightened levels of attention, expectancy, or cognitive processing linked to uncertain reward results [83]. These results confirmed the hypothesis of the RL theory regarding the FRN amplitude, depending on the relationship between actual and expected outcomes [12,13,16,20,22,82]. Moreover, this may be due to this difference in saliency of the size of losses. loss aversion, a cornerstone of behavioral economics, posits that losses loom larger than equivalent gains. Tversky and Kahneman (1991) [84] famously demonstrated that losses are roughly twice as psychologically potent as gains. When faced with choices involving equal magnitude wins and losses, this asymmetry becomes pronounced. For instance, the prospect of losing $100 is often more distressing than the joy of gaining the same amount.

This particular finding was exclusively pronounced among participants with high levels of dominance, who exhibited a more negative RPE amplitude than those with low dominance in the UNEXP-uncertain condition. Previous studies showed that activity in the ventral striatum, an area associated with reward processing, is higher when participants receive unexpected rewards compared to expected rewards. This finding provided evidence for dopamine-dependent PEs, which occur when the actual outcome of an event differs from the expected outcome [85,86]. It has been observed that high

dominance may be associated with greater reward sensitivity, and dominance may have influenced the neural processing of rewards and behavior related to rewards [87]. Interestingly, individual differences in reward sensitivity were found to be related to differences in the magnitude of the PE signal in the ventral striatum. Specifically, individuals who were more sensitive to rewards showed a stronger PE signal in response to unexpected rewards compared to those who were less sensitive to rewards [85].

In conjunction with the observed differences in FRN latency (shorter FRN latency for negative than positive feedback in the low-dominance group) were explained in terms of the brain's heightened sensitivity and responsiveness to negative outcomes It may have reflected the brain's evolutionary adaptation to prioritize detecting and processing potential threats or errors to facilitate learning and adaptive behavior [7]. As was previously mentioned, the increased sensitivity to negative feedback in individuals with low dominance may have been attributed to their heightened sensitivity to perceived threats or their tendency to concentrate on negative information [46]. This could have resulted in heightened vigilance and reactivity to negative stimuli.

**4.2.2 P3.** Regarding P3 modulation, high-dominance individuals exhibited significantly larger P3 amplitudes compared to their low-dominance counterparts. Furthermore, P3 amplitude was influenced by both feedback valence and condition, with more pronounced responses observed for positive feedback and unexpected uncertain outcomes. Previous research has suggested that power is associated with better access to rewards and a stronger bias to respond to reward-related cues [51]. P3 is a top-down controlled process that is sensitive to the evaluation of outcomes. Within the range of this potential, factors related to the allocation of attentional resources, such as reward capacity, reward magnitude, and expectation of reward size, also played a role [13]. Wu and Zhou (2009) [13] suggested that the amplitude of P3 could vary depending on the emotional valence of a stimulus. Still, this effect may have been modulated by the expectation of the stimulus. Specifically, they proposed that P3 variation due to valence depended on the expectations being set and then violated. However, in expected-certain trials, where the outcome was already known, there may have been no violation of expectations, which may explain why the valence effect was not as strong in these trials in the study. Therefore, it is believed that the P3 components reflected different aspects of feedback processing.

In general, the evidence showed that people with high social status were more focused on outcomes [46,88]. Additionally, P3 is related to various aspects of outcome evaluation [13]. Individuals with high social hierarchies have been found to possess greater attentional resources and allocate them more efficiently to stimuli related to their social status and power. These individuals were associated with a stronger tendency to respond to reward-related cues, which was likely related to their ability to obtain rewards and approach behaviors [43]. Boksem, et al. (2012) [49] demonstrated that the feeling of power triggered the brain's motivational mechanisms responsible for regulating approach behavior. Specifically, high-status individuals were associated with a stronger tendency to respond to reward-related cues. This means that individuals with higher social status positions may have been more likely to seek out and engage with stimuli associated with positive outcomes or rewards. This bias towards responding to reward-related cues may have been related to the experience of power, as previous research had shown that feeling powerful could activate the brain's motivational systems responsible for regulating approach behavior. Additionally, the heightened sensitivity to reward-related cues among high-status individuals could have contributed to their ability to maintain their social status and achieve success in their pursuits [88].

Also, the unexpected-uncertain condition showed the longest latency compared to other conditions, particularly in negative feedback, indicating that stimulus assessment was challenging in these situations. Consistent with stimulus assessment processes, the P3 latency was prolonged when the features of the stimulus or distractor were ambiguous [89]. Yeung et al. (2005) [14] observed that the P3 latency was longer for negative than positive feedback, suggesting that negative valence delayed the stimulus evaluation process. The study concluded that the combination of low probability and negative valence significantly influenced the stimulus evaluation and delayed it via top-down processes. The level of arousal can influence the processing system and determine the amount of attention available for task performance. The

P3 amplitude was believed to reflect attentional resources when task demands were low, resulting in a relatively large amplitude and a short peak latency. However, when tasks required more attentional resources, the P3 amplitude was smaller, and the peak latency was longer due to the allocation of processing resources to the task [72].

It's worthwhile to mention, the mismatch between the neural group differences and the absence of a main behavioral effect is worth deeper reflection. Interestingly, this kind of neural-behavioral dissociation isn't unique to our study. In previous works, such as those by [93], this pattern was also observed, where no significant difference was found in behavioral observations between the high and low dominance groups, whereas electrophysiological data showed a clear difference. Also, it mirrors patterns seen in other areas, such as research on feedback processing in substance-use populations, where neural markers like FRN/P3 amplitude or medial-frontal theta clearly distinguish groups even when their overall accuracy is the same (e.g., [17]). One possible explanation is that this dissociation reflects the inherently complex, multi-stage nature of cognitive processing. Neural signals are often sensitive to subtle shifts in attention, evaluation, or learning that may not be immediately apparent in overt behavior. In other words, just because the brain is reacting differently doesn't always mean we'll see a clear change in how someone responds—at least not right away. This gap between neural and behavioral measures may stem from the different temporal and mechanistic properties of the brain versus behavior. Neural activity often captures earlier stages of processing—like attention allocation or stimulus evaluation—long before a button is pressed or a response is made. It's entirely possible for two groups to show distinct neural patterns even if their outward behavior (e.g., RTs or accuracy) ends up looking similar. For example, early neural markers often reflect the very first stages of perception and evaluation—when you're just getting a sense of what's happening—long before you press a button. Those initial processes could vary significantly between high- and low-dominance individuals, even if, by the time the motor command goes out, everyone ends up behaving the same. Importantly, the motor stage introduces another layer of variability. Even after a decision is formed, it still has to be translated into an action. This step can be influenced by things like motor readiness, task strategy, or even fatigue—all of which can blur group-level effects that originated earlier in the process. As a result, differences in neural signatures of learning or evaluation (like variability or timing) may not always align neatly with average behavioral outputs [90–92,94].

Moreover, the large number of trials in our task could also contribute to this pattern. Practice-based explanation is that extensive practice across many trials allows behavior to converge over time, potentially masking group differences that were more apparent early on. Our exploratory time-course analyses support this idea: Specifically, descriptive trends suggested that participants with high social dominance appeared to show quicker RT improvements and more stable performance early in the task, while those with lower dominance gradually caught up. However, by the end of training, these transient differences had diminished, resulting in no reliable group effect when performance was averaged across the full task.

The correlation findings reveal that individual differences in dominance, as indexed by PRF-d scores, systematically modulate neural sensitivity to feedback under varying expectancy and uncertainty. Across both certain and uncertain contexts, lower dominance individuals were associated with more pronounced (i.e., more negative) FRN amplitudes, indicating that individuals with reduced feelings of power become especially vigilant to negative or unexpected outcomes. This effect was strongest in the unexpected–uncertain negative condition, suggesting that low-dominance participants are particularly sensitive to unpredictable adverse feedback [46].

Moreover, higher dominance people were characterized by higher effect amplitudes of P3, again only in the unexpected–uncertain negative condition. Because P3 is most often related to the motivational significance of stimuli and the allocation of cognitive resources, this pattern may suggest that dominant individuals might engage in extra evaluative or regulatory processes in the face of unexpected negative feedback [13,46,88]. The positive relationship between PRF-d scores and FRN-derived RPEs in the same unexpected–uncertain negative context underscores that higher-dominance individuals experience relatively larger RPEs—reflecting a mismatch between expected and actual outcomes—when outcomes are both negative and unpredictable. Together, these results suggest a dual mechanism by which dominance

amplifies early evaluative processing (FRN) and later motivational updating (P3 and RPE) in response to unanticipated negative events, highlighting the role of social power perceptions in shaping the neural dynamics of feedback processing.

Finally, our observations can profitably be interpreted in line with RL theory, which postulates that behavior is shaped by PEs (i.e., the difference between expected and actual outcome) that update the value attached to choice options through a learning rate parameter [95,96]. In this framework, FRN is widely considered to indicate a signed RPE, while the P3 component is thought to reflect the salience or associability of outcomes, scaling with the absolute magnitude of the PE [7,52]. In our study, both high- and low-dominance groups ultimately exhibited comparable behavioral learning trajectories during training—accuracy improvements and decreasing RTs—indicating similar effective learning rates across groups. A reflection of this is the pattern of the FRN amplitude, which was modulated by valence and expectancy of the outcome but did not differ between groups, implying matched computations of RPEs across levels of social dominance. However, the high-dominance group showed larger P3 amplitudes, especially with unexpected positive feedback, suggesting that there may be increased attentional engagement or salience attribution to rewarding outcomes. This dissociation matches RL models that argue for a separable role of RPE computation and outcome integration and suggests that dominance may modulate downstream processes related to outcome evaluation, rather than the very core error-driven learning mechanism.

## 4.3 Limitation

We calculated RPE by subtracting positive feedback from negative feedback. While we have successfully drawn from prior literature (e.g., [12,22,50–54], we acknowledge that this approach might lack precision and may not discern which specific ERP or experimental condition was influenced. Single-trial modeling approaches have significantly contributed to clarifying theoretical discussions in the context of reward processing within EEG signals [97].

While the present study employed ERP analyses to investigate the neural correlates of social dominance and decision-making under uncertain conditions, future research could benefit from utilizing other analyses, such as time-frequency and power spectral analysis techniques. These methods offer a more nuanced examination of oscillatory brain activity, potentially revealing additional insights into the dynamic neural processes underlying these cognitive functions. By exploring these alternative analytical approaches, researchers can further elucidate the temporal and spectral characteristics of brain signals associated with social dominance and decision-making.

Furthermore, owing to COVID-19 pandemic-related laboratory constraints, the sample was confined to university students, potentially limiting the generalizability of the findings to other groups. In order to enhance the research scope, future studies should incorporate a more diverse and representative sample, allowing for broader insights and increasing the generalizability of the findings.

Although we have not actually fitted RL models here, the behavioral and ERP results could be used to support or refute hypotheses in future modeling studies. For instance, extending this interpretation further, one might fit a Q-learning or RL-DDM model (reinforcement learning drift diffusion model) to choice and ERP data to estimate separate learning rates for gains and losses.

Another limitation of the present study is that the task was not optimized to directly assess attentional mechanisms. Although the neural findings suggest group differences in attentional engagement, our behavioral indices (accuracy, RT) may not have sensitivity comparable to neural measures and thus failed to reflect such effects. Future work could employ paradigms specifically designed to probe attention, such as Posner cueing or sustained attention tasks or tasks that are more sensitive to attention, in combination with computational modeling approaches like the DDM, where drift rate provides a quantitative marker of attentional allocation and evidence accumulation [8,98–102]. Such approaches would allow a more precise test of whether social dominance modulates reinforcement learning via attentional mechanisms, thereby extending the neural–behavioral dissociation reported here.

## 5. Conclusions

In conclusion, this study aimed to explore the impact of social dominance on feedback processing during decision-making under uncertainty. The findings revealed that individuals with lower social dominance displayed larger FRN amplitudes, indicating a greater concern about their performance evaluation. Conversely, individuals with high social dominance exhibited larger P3 amplitudes, suggesting that they allocate attentional resources more efficiently. Higher social dominance was associated with a pronounced FRN amplitude difference after exposure to unexpected-uncertain conditions. Low-dominance individuals exhibited shorter FRN latency for negative feedback. The P3 component displayed larger amplitudes following unexpected-uncertain conditions, particularly in response to positive feedback, and longer latency in unexpected-certain positive conditions.

These results suggest that social dominance traits play a crucial role in feedback processing under uncertain conditions. They highlight the importance of considering individual differences in social dominance when examining these processes. The study contributes to the existing literature on the interplay between social dominance and uncertainty in decision-making. Further studies in this direction can provide a more comprehensive understanding of the complex factors that influence human decision-making.

## Supporting information

**S1 File. PRF-d distribution.**
(ZIP)

## Acknowledgments

Our gratitude extends to the participants who contributed to this experiment, and all the individuals who played a role in its success.

## Author contributions

**Conceptualization:** Soomaayeh Heysieattalab.

**Formal analysis:** Saeedeh Khosravi, Lydia kogler.

**Funding acquisition:** Soomaayeh Heysieattalab.

**Investigation:** Saeedeh Khosravi.

**Methodology:** Lydia kogler.

**Project administration:** Lydia kogler.

**Resources:** Reza Khosrowabadi, Touraj Hashemi.

**Supervision:** Birgit Derntl, Soomaayeh Heysieattalab.

**Validation:** Lydia kogler, Birgit Derntl, Soomaayeh Heysieattalab.

**Writing – original draft:** Saeedeh Khosravi, Birgit Derntl.

**Writing – review & editing:** Lydia kogler, Soomaayeh Heysieattalab.

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
