## [Decision Letter · Decision Letter 0]

7 Apr 2025

PONE-D-24-52452The interplay between social dominance and decision-making under expected and unexpected uncertainty: Evidence from event-related potentialsPLOS ONE

Dear Dr. Heysieattalab,

Thank you for submitting your manuscript to PLOS ONE. After careful consideration, we feel that it has merit but does not fully meet PLOS ONE’s publication criteria as it currently stands. Therefore, we invite you to submit a revised version of the manuscript that addresses the points raised during the review process.

I've reviewed the manuscript and agree with the concerns raised by the two reviewers regarding both the behavioral and neural data. The authors need to address these points thoroughly in their revision.

A critical improvement needed is the clear articulation of hypotheses in both behavioral and neural terms. Each hypothesis should be presented in a testable manner that establishes clear predictions. This clarity will significantly help readers understand the relationship between the behavioral findings and neural data patterns.

The manuscript would also benefit from stronger theoretical framing. Several ERP components discussed are traditionally explained within reinforcement learning frameworks, yet this theoretical perspective is underdeveloped in the current version. I recommend expanding the theoretical discussion to situate your findings within established models of reinforcement learning and to highlight the unique contributions your work makes to this literature.

These revisions will strengthen both the empirical foundation and theoretical significance of the work.

We look forward to receiving your revised manuscript.

Kind regards,

Rei Akaishi

Academic Editor

PLOS ONE

Journal Requirements:

“This project holds a scholarship entitled “TabrizU-300” by the International Academic Cooperation Directorate University of Tabriz, Iran.”

3. In this instance it seems there may be acceptable restrictions in place that prevent the public sharing of your minimal data. However, in line with our goal of ensuring long-term data availability to all interested researchers, PLOS’ Data Policy states that authors cannot be the sole named individuals responsible for ensuring data access (http://journals.plos.org/plosone/s/data-availability#loc-acceptable-data-sharing-methods).

Additional Editor Comments:

I've reviewed the manuscript and agree with the concerns raised by the two reviewers regarding both the behavioral and neural data. The authors need to address these points thoroughly in their revision.

A critical improvement needed is the clear articulation of hypotheses in both behavioral and neural terms. Each hypothesis should be presented in a testable manner that establishes clear predictions. This clarity will significantly help readers understand the relationship between the behavioral findings and neural data patterns.

The manuscript would also benefit from stronger theoretical framing. Several ERP components discussed are traditionally explained within reinforcement learning frameworks, yet this theoretical perspective is underdeveloped in the current version. I recommend expanding the theoretical discussion to situate your findings within established models of reinforcement learning and to highlight the unique contributions your work makes to this literature.

These revisions will strengthen both the empirical foundation and theoretical significance of the work.

Reviewers' comments:

Reviewer's Responses to Questions

**Comments to the Author**

1. Is the manuscript technically sound, and do the data support the conclusions?

Reviewer #1: Partly

Reviewer #2: Yes

2. Has the statistical analysis been performed appropriately and rigorously? 

Reviewer #1: Yes

Reviewer #2: Yes

3. Have the authors made all data underlying the findings in their manuscript fully available?

Reviewer #1: No

Reviewer #2: No

4. Is the manuscript presented in an intelligible fashion and written in standard English?

Reviewer #1: Yes

Reviewer #2: Yes

5. Review Comments to the Author

Reviewer #1: This study aimed to unravel the neural and behavioral patterns associated with decision-making across varying social dominance levels under conditions of uncertainty. They investigated the properties of key

neural correlates of feedback processing, including feedback-related negativity (FRN), and P3 components, and reward prediction error (RPE) signals. There are some suggestions.

The basis of the hypothesis is not clear, please revise the hypotheses parts, e.g. line 147“We expected that people with lower levels of dominance show more sensitivity to performance appraisal and more strongly evaluate their own performance”why did you get this hypothesis? and what is "more strongly evaluate their own performance"?

Line 96-126 is too long, please split it.

In the training session�it included only 60 trials,How can you be sure that these trials have led to subjects learning the characteristics of each cue?

In the experimental session, the subjects will adjust their expectations according to the various results brought by the cue. The influence of the previous association in the training session may be effective only at the beginning, as the number of formal trials increases, the effect of this expectation will become less and less. How many of the 500 trials are affected by the expectation learnt in the training session? It is suggested that the researchers should analyze the whole experiment in sections and examine the characteristics of individual learning process.

Why only three conditions in the experiment? 1) expected-certain feedback, 2) expected-uncertain feedback, and 3) unexpected uncertain feedback, Why not design unexpected certain conditions

It could be a very important result to test the correlation among personality scores, brain electricity, and behavior to investigate the explanation path of personality-brain mechanism-behavior.

What are the different meanings of amplitude, latency and RPE of ERP components?The implication that independent variables affect these three dependent variable indicators should be explained in the discussion section.

Reviewer #2: This study investigates the relationship between social dominance and different types of uncertainty, both at the behavioral and neural levels. The study is well-designed, and the analyses are well-executed. Below are my main and minor comments:

Main comments:

Since the authors use a learning task, I believe it would be helpful to examine the learning curves both preference and reaction times (RTs), “during” each block, and separately for each cue. Given the large number of trials per condition, it is possible that behavioral effects differ substantially between the early and late phases of each block. Furthermore, because this is a learning task, the authors are in a strong position to investigate whether individuals with different levels of social dominance learn differently. Therefore, I recommend analyzing behavioral data dynamically across time, not just as grand averages. The author can effectively quantify “learning rates” in their data, based on the different experimental conditions and participant groups.

Again, the task used in this study is a reward-based learning paradigm, which is well-studied in the reinforcement learning literature. Since here there are known neural proxies for reward prediction error (e.g., FRN, P3), it would be valuable to relate the current neural findings to RL theory. Ideally, fitting a simple RL model to the behavioral data could reveal whether learning mechanisms differ by social dominance. I understand that this may be outside the scope of the current study, and I do not consider it essential. However, at a minimum, a paragraph in the discussion exploring how the findings could be interpreted through the lens of RL theory would be very helpful. especially in light of Comment 1, which may shed light on whether social dominance is related to differences in learning processes (specifically learning rate).

This study reports a neural effect of group (dominance), but no main behavioral effect; only a group × condition interaction. I think it is important to discuss this discrepancy more directly. What might contribute to the presence of group differences at the neural level, but not behaviorally? One possibility is related to Comment 1: with a large number of trials per condition, behavioral responses may converge over time, masking early-stage group differences. Examining behavior during early phases of the task may help clarify this. I don’t view this discrepancy as a major issue, but I believe the paper would benefit from a thoughtful discussion of it.

Minor comments:

I found the description of how dominance is determined based on the PRF_d scores a bit unclear. The text mentions that extreme values (0 and 16) were extracted using data from the remaining 322 subjects. However, the authors report the mean PRF_d scores for both groups, and these values are not exactly 0 or 16. Clarification on this criterion would be helpful.

Do we have access to the distribution of PRF_d scores across the 322 subjects? It would be useful to see what proportion of the population was selected and based on what distribution. This would provide a clearer picture of how the authors defined dominance relative to the general population.

Additionally, I believe it would be helpful if the authors reported the percentage of trials, ICA components, and electrodes that were removed or interpolated. Including this information in the methods section would improve transparency and reproducibility.

6. PLOS authors have the option to publish the peer review history of their article (what does this mean? ). If published, this will include your full peer review and any attached files.

**Do you want your identity to be public for this peer review?** For information about this choice, including consent withdrawal, please see our Privacy Policy .

Reviewer #1: No

Reviewer #2: No

---

## [Author Response · Author response to Decision Letter 1]

18 Jul 2025

Author's Response to Editor and Reviewer Comments

We are grateful to the reviewers for dedicating their time and attention to our manuscript. We believe that the manuscript has notably improved following the revisions made in response to their comments. In this document, original reviewer comments are presented in standard font, and dark blue color, while our responses are shown in italics, and black. Where applicable, modifications to the text are highlighted in bold.

I've reviewed the manuscript and agree with the concerns raised by the two reviewers regarding both the behavioral and neural data. The authors need to address these points thoroughly in their revision.

A critical improvement needed is the clear articulation of hypotheses in both behavioral and neural terms. Each hypothesis should be presented in a testable manner that establishes clear predictions. This clarity will significantly help readers understand the relationship between the behavioral findings and neural data patterns.

The manuscript would also benefit from stronger theoretical framing. Several ERP components discussed are traditionally explained within reinforcement learning frameworks, yet this theoretical perspective is underdeveloped in the current version. I recommend expanding the theoretical discussion to situate your findings within established models of reinforcement learning and to highlight the unique contributions your work makes to this literature.

These revisions will strengthen both the empirical foundation and theoretical significance of the work.

First of all, thank you for the comments. Please find our answers to your insightful comments below:

Editor:

Answer: Thank you for the reminder. I have reviewed the PLOS ONE style templates and updated my files to ensure that all formatting, headings, and file names comply with PLOS ONE style requirements.

“This project holds a scholarship entitled “TabrizU-300” by the International Academic Cooperation Directorate University of Tabriz, Iran.”

Answer: Thank you for your constructive feedback on the financial disclosure statement. We have revised the declaration on the role of the funders as follows:

Funding: This project holds a scholarship entitled “TabrizU-300” by the International Academic Cooperation Directorate University of Tabriz, Iran. The funders had no role in study design, data collection and analysis, decision to publish, or preparation of the manuscript.

3. In this instance it seems there may be acceptable restrictions in place that prevent the public sharing of your minimal data. However, in line with our goal of ensuring long-term data availability to all interested researchers, PLOS’ Data Policy states that authors cannot be the sole named individuals responsible for ensuring data access (http://journals.plos.org/plosone/s/data-availability#loc-acceptable-data-sharing-methods).

Answer: Thank you for your valuable feedback and for highlighting the importance of long-term data access. In response, we have now uploaded all the required datasets underlying our findings to the Open Science Framework (OSF), a trusted and durable data repository that ensures persistent availability. The dataset can be accessed via the following link: “https://osf.io/4rsmq/”

We have updated the Data Availability statement in the revised manuscript accordingly and ensured that the data is publicly accessible through this platform. We believe this solution aligns with PLOS’s data sharing policy and provides long-term access for interested researchers.

page 44, Data availability: Data will be provided via the following link https://osf.io/4rsmg/.

Answer: Thank you for the suggestion. We have added captions for all Supporting Information files at the end of the manuscript and updated the corresponding in-text citations to match.

Page 56, Supporting information:

S1 Fig. PRF-d score distribution

S2 Fig. FRN component in Cz electrode. a) Grand averaged ERP waveforms at Fz electrode for low and high dominance groups in positive, and negative feedback in three conditions (EXP-certain, EXP-uncertain, and UNEXP-uncertain). b) topographical scalp for the difference waves FRN component (200-300 ms post-feedback window) for each of the positive, and negative conditions for high and low dominance groups; in μV. C) FRN means amplitude differences; Error bars denote SD. In all comparisons, "**" denotes p < 0.01, and "*" indicates p < .05 in all comparisons.

S3 Fig. FRN component in FCz electrode. a) Grand averaged ERP waveforms at Fz electrode for low and high dominance groups in positive, and negative feedback in three conditions (EXP-certain, EXP-uncertain, and UNEXP-uncertain). b) topographical scalp for the difference waves FRN component (200-300 ms post-feedback window) for each of the positive, and negative conditions for high and low dominance groups; in μV. C) FRN means amplitude differences; Error bars denote SD. In all comparisons, "**" denotes p < 0.01, and "*" indicates p < .05 in all comparisons.

Additional Editor Comments:

I've reviewed the manuscript and agree with the concerns raised by the two reviewers regarding both the behavioral and neural data. The authors need to address these points thoroughly in their revision.

A critical improvement needed is the clear articulation of hypotheses in both behavioral and neural terms. Each hypothesis should be presented in a testable manner that establishes clear predictions. This clarity will significantly help readers understand the relationship between the behavioral findings and neural data patterns.

Answer: Thank you very much for your insightful feedback. We have edited the manuscript accordingly to clearly outline our hypotheses in behavioral and neural terms. Each hypothesis is now explicitly stated, with clear testable predictions about the modulation of behavioral sensitivity to feedback and neural response (i.e., FRN and P3 amplitudes) by social dominance under varying conditions of expectancy and uncertainty. This revision enhances the clarity of the relationship between our behavioral findings and neural data patterns, thereby facilitating better comprehension for readers.

Page 7-8, 1. Introduction, lines 153-174: The main question of this study was whether the amplitude and latency of the FRN and P3 components differ between high-dominance and low-dominance individuals when decision outcomes are either expected or unexpected. Using a well-established decision-making paradigm (12, 22), participants received positive and negative feedback under three feedback conditions—expected-certain, expected-uncertain, and unexpected-uncertain—while EEG recordings captured FRN and P3 activity. Based on prior research indicating that individuals with lower social dominance are more attuned to social evaluation and exhibit heightened sensitivity to external judgments in uncertain or hierarchical contexts (46, 49), we hypothesized that they would show more pronounced behavioral responses to negative feedback—particularly when outcomes are unexpected and uncertain—as a reflection of increased performance monitoring and concern for evaluation. At the neural level, we predicted that low-dominance individuals would exhibit larger FRN amplitudes in response to negative feedback under unexpected-uncertain conditions, indicating greater sensitivity to negative PEs. In contrast, we expected P3 amplitudes to be generally higher in response to unexpected-uncertain feedback than expected feedback across all participants, with high-dominance individuals showing larger P3 amplitudes overall due to their stronger motivational engagement and greater sensitivity to reward salience. Furthermore, to isolate neural correlates of RPE, we computed valence difference waves (negative minus positive feedback) for both FRN and P3 components (12, 22, 50-54). Given the evidence that high-dominance individuals exhibit stronger responses to reward cues and are more driven by reward-seeking behavior (49, 55-59), we expected more pronounced negative RPE signals in the unexpected-uncertain condition among high-dominance individuals relative to their low-dominance counterparts.

The manuscript would also benefit from stronger theoretical framing. Several ERP components discussed are traditionally explained within reinforcement learning frameworks, yet this theoretical perspective is underdeveloped in the current version. I recommend expanding the theoretical discussion to situate your findings within established models of reinforcement learning and to highlight the unique contributions your work makes to this literature.

Answer: Thank you for your thoughtful and constructive feedback. We appreciate your suggestion to strengthen the theoretical framing. We expanded the introduction and discussion (At the comment of the second reviewer), and limitation in the manuscript to better highlight how our work contributes to this literature.

Page 3-8, 1. Introduction: Humans face a tradeoff between exploration and exploitation when making decisions, which influences behavior from daily activities to occupational choices (1). Decision-making is a complex and challenging process, and the outcomes of decisions are often unclear and may lead to unfavorable results, requiring individuals to be mindful of uncertainty while making choices (2, 3). Uncertainty in decision-making comes in two forms: expected and unexpected. Expected uncertainty is the familiar unreliability of actions with probabilistic outcomes, like flipping a coin. Unexpected uncertainty, on the other hand, is a surprise – a sudden change in a previously reliable association, like a lever that always gave a reward but now doesn't. In other words, unexpected uncertainty arises from a violation of strong action-outcome links, while expected uncertainty stems from violations of weak, probabilistic ones (4).

In contexts characterized by uncertainty, efficient decision-making is crucial to minimize harm and optimize well-being and prompt assessment of the consequences of actions holds significant importance (5). Individuals heavily rely on feedback—whether positive or negative—to guide their future behavior. The brain likely employs specialized mechanisms to assess outcome value, magnitude, and to link feedback information with mental importance and motivation (6). This cognitive process is facilitated by distinct systems in the brain dedicated to processing rewards and losses (5).

Theoretical frameworks pertaining to reinforcement learning (RL) furnish a valuable paradigm for comprehending these neural mechanisms. RL theory asserts that learning occurs through iterative experimentation, wherein individuals assess the repercussions of their behaviors—actions yielding outcomes that exceed expectations are reinforced, whereas those resulting in outcomes that fall short of expectations are diminished. Central to this process is the concept of the prediction error (PE), which is the discrepancy between expected and actual outcomes, driving internal learning signals and expectation updates (7, 8). One of the electroencephalographic components, known as the feedback-related negativity (FRN), is a well-established neural marker associated with loss processing (9). This negative deflection is typically observed 200-300 millisecond post-feedback onset in the frontal region, exhibits larger amplitudes in response to negative feedback compared to positive feedback (7, 9-16). Beyond simple outcome valence, the FRN is sensitive to factors such as uncertainty and unexpectedness (12). Research suggests that the FRN reflects the encoding of PEs, aligning with RL theory (16).

Within the RL framework, the FRN is understood to reflect a rapid negative PE signal—often tied to mesencephalic dopamine signals—relayed to the anterior cingulate cortex (ACC), where it supports behavioral adaptation by identifying outcomes that are worse than predicted. The salience of this signal increases when outcomes are surprising or when uncertainty is high, making FRN a core index of adaptive learning (7, 17). Studies by Yu et al. (2011) (18) and Pfabigan et al. (2011) (12) have shown that the FRN range is modulated by these factors, particularly for unexpected and uncertain outcomes. This suggests that the FRN may categorize outcomes as "good" or "bad" and signal "worse-than-expected" results via reward prediction error (RPE) signals. RPE represents the discrepancy between expected and actual outcomes, driving internal expectation updates (11, 19-23). The amplitude of RPE signals is influenced by outcome expectation, determined by factors such as certainty and likelihood. Mushtaq et al. (2011) (24)[22] defines certainty as the ability to accurately predict future events.

The P3 component plays a significant role in outcome evaluation and reward processing (25, 26). Following the detection of a mismatch between expected and actual results, the prediction model is updated to enhance its accuracy for future feedback. In response to reward and punishment stimuli, a positive deflection, known as P3, appears between 300 to 500 milliseconds post-feedback and is linked to this mechanism (27). Beyond its initial sensitivity to stimulus meaning and probability, the P3 component of the event-related potential (ERP) is essential in higher-order cognitive processes. It contributes to decision-making (27) and outcome evaluation (13), dynamically adapting its response to assess the functional significance of feedback stimuli (10, 28). This goes beyond mere performance tracking, reflecting the subjective significance and surprise associated with the stimuli (12, 29).

In RL terms, the P3 component is thought to reflect the allocation of attentional resources and motivational significance to feedback—particularly rewarding feedback. While the FRN signals an initial, automatic PE, the P3 is more involved in the consolidation and evaluation of the feedback's affective and motivational impact. Together, these components index distinct but complementary aspects of the RL feedback loop: immediate error detection (FRN) and the higher-order appraisal of

---

## [Decision Letter · Decision Letter 1]

19 Aug 2025

PONE-D-24-52452R1The interplay between social dominance and decision-making under expected and unexpected uncertainty: Evidence from event-related potentialsPLOS ONE

Dear Dr. Heysieattalab,

Thank you for submitting your manuscript to PLOS ONE. After careful consideration, we feel that it has merit but does not fully meet PLOS ONE’s publication criteria as it currently stands. Therefore, we invite you to submit a revised version of the manuscript that addresses the points raised during the review process.

The authors have generally addressed the points raised by both the reviewers and the editors. The responses are satisfactory overall, though they are presented in lengthy passages copied directly from the revised manuscript. It would strengthen the response if the authors provided more concise, direct answers to the specific questions raised.

Additionally, the data repository requires attention. First, the stated link (https://osf.io/4rsmg/) is inactive. Second, the repository lacks a metadata file (e.g., readme.txt) that describes its contents. Such a file should be included for clarity and transparency. Third, the repository currently contains only summary data. If feasible, the authors are encouraged to also upload the raw data to enhance reproducibility and data integrity.

The authors should also address the remaining comment of the one reviewer in the second round.

We look forward to receiving your revised manuscript.

Kind regards,

Rei Akaishi

Academic Editor

PLOS ONE

Journal Requirements:

Additional Editor Comments :

The authors have generally addressed the points raised by both the reviewers and the editors. The responses are satisfactory overall, though they are presented in lengthy passages copied directly from the revised manuscript. It would strengthen the response if the authors provided more concise, direct answers to the specific questions raised.

Additionally, the data repository requires attention. First, the stated link (https://osf.io/4rsmg/) is inactive. Second, the repository lacks a metadata file (e.g., readme.txt) that describes its contents. Such a file should be included for clarity and transparency. Third, the repository currently contains only summary data. If feasible, the authors are encouraged to also upload the raw data to enhance reproducibility and data integrity.

The authors should also address the remaining comment of the one reviewer in the second round.

Reviewers' comments:

Reviewer's Responses to Questions

**Comments to the Author**

1. If the authors have adequately addressed your comments raised in a previous round of review and you feel that this manuscript is now acceptable for publication, you may indicate that here to bypass the “Comments to the Author” section, enter your conflict of interest statement in the “Confidential to Editor” section, and submit your "Accept" recommendation.

Reviewer #2: All comments have been addressed

2. Is the manuscript technically sound, and do the data support the conclusions?

Reviewer #2: Yes

3. Has the statistical analysis been performed appropriately and rigorously? 

Reviewer #2: Yes

4. Have the authors made all data underlying the findings in their manuscript fully available?

Reviewer #2: Yes

5. Is the manuscript presented in an intelligible fashion and written in standard English?

Reviewer #2: Yes

6. Review Comments to the Author

Reviewer #2: Thanks to the authors for providing a revised version. I believe they have done a great job in addressing the concerns. Here, I am pointing out some minor issues that I suggest should be addressed.

(1) Take a look at these two sentences:

participants with high social dominance showed quicker RT

improvements and more stable performance early in the task, while those with lower dominance

gradually caught up.

In our study, both high- and low-dominance groups exhibited comparable behavioral

learning trajectories during training—accuracy improvements and decreasing RTs—suggesting

similar effective learning rates.

I think this may be somewhat contradictory. Based on the reported statistics, it appears that there is no effect of group in learning, which is interesting. However, this could have been communicated more clearly.

(2) I think the authors argue in several places in the manuscript that the discrepancy between neural and behavioral data could be due to variables that are more easily detectable in neural data but less so in behavioral data. I agree with this interpretation. One proposed candidate is attention. The neural data equivalent of RL models would also point to a similar candidate. In many perceptual decision-making tasks, however, attention is both behaviorally and computationally measurable, for example through differences in reaction time and/or accuracy, and through drift rate in the drift diffusion model (DDM). Overall, I think this study suggests that attention differs between dominance groups. This is both experimentally and computationally testable via tasks that are more sensetive to attention change than the current task. Therefore, I suggest that the authors make this point critically clear and encourage readers to test the findings of the current work in future studies.

(3) This sentence has been appended to the end of a paragraph related to learning, whereas i think it should have been appended to the one discussing the discrepancy between behavioral and neural data.

Interestingly, this kind of neural-behavioral dissociation isn’t unique to our study. In previous

works, such as those by (94), this pattern was also observed, where no significant difference was found in behavioral observations between the high and low dominance groups, whereas

electrophysiological data showed a clear difference. Also, it mirrors patterns seen in other areas,

such as research on feedback processing in substance-use populations, where neural markers like FRN/P3 amplitude or medial-frontal theta clearly distinguish groups even when their overall

accuracy is the same (e.g., (17)).

7. PLOS authors have the option to publish the peer review history of their article (what does this mean? ). If published, this will include your full peer review and any attached files.

**Do you want your identity to be public for this peer review?** For information about this choice, including consent withdrawal, please see our Privacy Policy .

Reviewer #2: No

---

## [Author Response · Author response to Decision Letter 2]

4 Sep 2025

Response to Editor and reviewer:

We sincerely thank the editor and reviewers for the time and effort they invested in evaluating our manuscript. We believe that the revisions made in response to their feedback have substantially improved the quality of the work. In this document, the reviewers’ original comments are presented in standard font and dark blue, while our responses are provided in black italics. Where relevant, revisions to the manuscript are highlighted in bold.

Editor

the data repository requires attention. First, the stated link (https://osf.io/4rsmg/) is inactive. Second, the repository lacks a metadata file (e.g., readme.txt) that describes its contents. Such a file should be included for clarity and transparency. Third, the repository currently contains only summary data. If feasible, the authors are encouraged to also upload the raw data to enhance reproducibility and data integrity.

Answer: Thank you for the reminder, we acknowledge the issues with the OSF repository:

The previously provided link (https://osf.io/4rsmq/)is now corrected and active.

A `readme.txt` file has been added to describe the contents of the repository, including the descriptions, summary data and associated materials. In addition to summary, we have uploaded the behavioral and neural datasets, enhancing reproducibility and data transparency.

We believe these updates fully address the editor’s concerns regarding accessibility, metadata clarity, and reproducibility.

Reviewer #2:

Thanks to the authors for providing a revised version. I believe they have done a great job in addressing the concerns. Here, I am pointing out some minor issues that I suggest should be addressed.

Thank you very much for your positive valuation of our revised manuscript. We truly appreciate your constructive feedback and will carefully address the minor issues you have pointed out.

(1) Take a look at these two sentences:

participants with high social dominance showed quicker RT improvements and more stable performance early in the task, while those with lower dominance gradually caught up.

In our study, both high- and low-dominance groups exhibited comparable behavioral learning trajectories during training—accuracy improvements and decreasing RTs—suggesting

similar effective learning rates.

I think this may be somewhat contradictory. Based on the reported statistics, it appears that there is no effect of group in learning, which is interesting. However, this could have been communicated more clearly.

Answer: We appreciate the reviewer’s careful reading and insightful comment. We agree that the way these two sentences were phrased could create the impression of a contradiction. Our intent was to distinguish between two different levels of analysis:

Exploratory time-course analysis (early phase): These analyses suggested that high-dominance participants appeared to show quicker RT improvements and more stable performance during the initial stages of training, whereas low-dominance participants caught up gradually. This was meant to highlight transient trends rather than group-level differences that persisted across the full task.

Confirmatory statistical analysis (whole trajectory): The mixed ANOVA and regression analyses showed no significant main effect of group and broadly comparable accuracy and RT improvements across dominance levels. In other words, though descriptive trends from the early part of an experiment suggested that perhaps the high-dominance group stabilized performance faster than their counterpart, differences in learning rates measured over the full course of training were not statistically distinguishable.

Therefore, to prevent confusion, we revised the text to make this distinction explicit.

Page 41: Moreover, the large number of trials in our task could also contribute to this pattern. Practice-based explanation is that extensive practice across many trials allows behavior to converge over time, potentially masking group differences that were more apparent early on. Our exploratory time-course analyses support this idea: Specifically, descriptive trends suggested that participants with high social dominance appeared to show quicker RT improvements and more stable performance early in the task, while those with lower dominance gradually caught up. However, by the end of training, these transient differences had diminished, resulting in no reliable group effect when performance was averaged across the full task.

Page 42: Finally, our observations can profitably be interpreted in line with RL theory, which postulates that behavior is shaped by PEs (i.e., the difference between expected and actual outcome) that update the value attached to choice options through a learning rate parameter (95, 96). In this framework, FRN is widely considered to indicate a signed RPE, while the P3 component is thought to reflect the salience or associability of outcomes, scaling with the absolute magnitude of the PE (7, 52). In our study, both high- and low-dominance groups ultimately exhibited comparable behavioral learning trajectories during training—accuracy improvements and decreasing RTs—indicating similar effective learning rates across groups.

(2) I think the authors argue in several places in the manuscript that the discrepancy between neural and behavioral data could be due to variables that are more easily detectable in neural data but less so in behavioral data. I agree with this interpretation. One proposed candidate is attention. The neural data equivalent of RL models would also point to a similar candidate. In many perceptual decision-making tasks, however, attention is both behaviorally and computationally measurable, for example through differences in reaction time and/or accuracy, and through drift rate in the drift diffusion model (DDM). Overall, I think this study suggests that attention differs between dominance groups. This is both experimentally and computationally testable via tasks that are more sensitive to attention change than the current task. Therefore, I suggest that the authors make this point critically clear and encourage readers to test the findings of the current work in future studies.

Answer: We thank the reviewer for this thoughtful comment and fully agree that attention is a strong candidate mechanism underlying the neural–behavioral dissociation we observed. As noted, while our task allowed us to identify group differences in FRN and P3 amplitudes, the behavioral measures in this paradigm (accuracy, RT) may not have been sensitive enough to capture delicate attentional effects. We appreciate the reviewer’s suggestion to highlight that attention can be experimentally and computationally measured in other paradigms (e.g., drift diffusion modeling of drift rate) and to encourage future work to test our findings using tasks optimized for attention. We have now clarified this in the limitations section, emphasizing attention as a promising avenue for further empirical and modeling-based investigation.

Page 44, limitation: Another limitation of the present study is that the task was not optimized to directly assess attentional mechanisms. Although the neural findings suggest group differences in attentional engagement, our behavioral indices (accuracy, RT) may not have sensitivity comparable to neural measures and thus failed to reflect such effects. Future work could employ paradigms specifically designed to probe attention, such as Posner cueing or sustained attention tasks or tasks that are more sensitive to attention, in combination with computational modeling approaches like the DDM, where drift rate provides a quantitative marker of attentional allocation and evidence accumulation (8, 98-102). Such approaches would allow a more precise test of whether social dominance modulates reinforcement learning via attentional mechanisms, thereby extending the neural–behavioral dissociation reported here.

(3) This sentence has been appended to the end of a paragraph related to learning, whereas I think it should have been appended to the one discussing the discrepancy between behavioral and neural data.

Interestingly, this kind of neural-behavioral dissociation isn’t unique to our study. In previous

works, such as those by (94), this pattern was also observed, where no significant difference was found in behavioral observations between the high and low dominance groups, whereas

electrophysiological data showed a clear difference. Also, it mirrors patterns seen in other areas,

such as research on feedback processing in substance-use populations, where neural markers like FRN/P3 amplitude or medial-frontal theta clearly distinguish groups even when their overall

accuracy is the same (e.g., (17)).

Answer: We appreciate the comment and agree that the sentence fits better with the discussion of the discrepancy between behavioral and neural data, and we've moved it accordingly.

Page 40: It's worthwhile to mention, the mismatch between the neural group differences and the absence of a main behavioral effect is worth deeper reflection. Interestingly, this kind of neural-behavioral dissociation isn’t unique to our study. In previous works, such as those by (93), this pattern was also observed, where no significant difference was found in behavioral observations between the high and low dominance groups, whereas electrophysiological data showed a clear difference. Also, it mirrors patterns seen in other areas, such as research on feedback processing in substance-use populations, where neural markers like FRN/P3 amplitude or medial-frontal theta clearly distinguish groups even when their overall accuracy is the same (e.g., (17)). One possible explanation is that this dissociation reflects the inherently complex, multi-stage nature of cognitive processing. Neural signals are often sensitive to subtle shifts in attention, evaluation, or learning that may not be immediately apparent in overt behavior.

---

## [Decision Letter · Decision Letter 2]

23 Sep 2025

The interplay between social dominance and decision-making under expected and unexpected uncertainty: Evidence from event-related potentials

PONE-D-24-52452R2

Dear Dr. Heysieattalab,

We’re pleased to inform you that your manuscript has been judged scientifically suitable for publication and will be formally accepted for publication once it meets all outstanding technical requirements.

Kind regards,

Rei Akaishi

Academic Editor

PLOS ONE

Additional Editor Comments (optional):

Thank you for your diligent work in addressing all the review comments. The manuscript is now ready for publication.

Reviewers' comments:

Reviewer's Responses to Questions

**Comments to the Author**

1. If the authors have adequately addressed your comments raised in a previous round of review and you feel that this manuscript is now acceptable for publication, you may indicate that here to bypass the “Comments to the Author” section, enter your conflict of interest statement in the “Confidential to Editor” section, and submit your "Accept" recommendation.

Reviewer #2: All comments have been addressed

2. Is the manuscript technically sound, and do the data support the conclusions?

Reviewer #2: Yes

3. Has the statistical analysis been performed appropriately and rigorously? 

Reviewer #2: Yes

4. Have the authors made all data underlying the findings in their manuscript fully available?

Reviewer #2: Yes

5. Is the manuscript presented in an intelligible fashion and written in standard English?

Reviewer #2: Yes

6. Review Comments to the Author

Reviewer #2: Dear Authors,

Thank you for carefully addressing the comments. I am satisfied with the revisions, and I believe the manuscript is now in good shape.

7. PLOS authors have the option to publish the peer review history of their article (what does this mean? ). If published, this will include your full peer review and any attached files.

**Do you want your identity to be public for this peer review?** For information about this choice, including consent withdrawal, please see our Privacy Policy .

Reviewer #2: No

---

## [Editor Report · Acceptance letter]

PONE-D-24-52452R2

PLOS ONE

Dear Dr. Heysieattalab,

I'm pleased to inform you that your manuscript has been deemed suitable for publication in PLOS ONE. Congratulations! Your manuscript is now being handed over to our production team.

Kind regards,

on behalf of

Dr. Rei Akaishi

Academic Editor

PLOS ONE